# StyleAT: Defending Face Recognition Against Semantic Attacks

## Abstract

With face-recognition models now embedded in everyday authentication and surveillance, recent works have pinpointed a critical weakness: these models remain acutely vulnerable to adversarial semantic edits. I.e., adversarially produced semantic alterations to the input, such as slight aging or pose changes, can induce misclassifications. Certain existing attacks are powerful, but they can be computationally costly, rendering them inadequate for developing defenses (e.g., through adversarial training). To fill the gap, we introduce **BoundStyle**, a potent semantic attack operating in StyleGAN's rich latent space to maximize misclassification rates. Notably, BoundStyle is significantly more efficient than equally powerful attacks, making it suitable for adversarial training. Building on BoundStyle, we develop **StyleAT**, an efficient adversarial training scheme that incorporates low-budget attack variants yet defends against stronger and unseen semantic attacks. We evaluate on two datasets unseen during training (LFW and VGG-Face) and five models, and find that StyleAT boosts robust accuracy against state-of-the-art attacks (DiffPrivate and BoundStyle) and outperforms common defenses (DOA and classical filters) in various settings.

## 1 Introduction

Face-recognition (FR) technologies are employed in various security-critical applications, including for surveillance and access control (Introna & Nissenbaum, 2009). Failures of such systems may be pernicious; for instance, false negatives may enable criminals to avoid surveillance, whereas false positives may provision unauthorized access to important resources protected by access control. However, unfortunately, similar to other machine-learning (ML) models that can be evaded by adversarial examples at inference time (Goodfellow et al., 2015; Szegedy et al., 2014), FR models are vulnerable to *general semantic attacks*—a class of adversarial example attacks that introduce slight semantic changes (e.g., addition of accessories or alterations of pose, expression, or age) to fool FR despite preserving the identity of the subject in the image (Barattin et al., 2023; Jia et al., 2022; Le & Carlsson, 2024; 2025; Liu et al., 2024).

Existing general semantic attacks mostly rely on generative models, such as StyleGAN (Le & Carlsson, 2024) and latent diffusion models (Le & Carlsson, 2025), to discover adversarial semantic edits through latent-space perturbations that mislead FR. However, these attacks suffer from certain limitations: Some attacks such as StyleAdv achieve *limited attack success* (Le & Carlsson, 2025); other attacks such as AMT-GAN produce edits with *low visual fidelity* (Le & Carlsson, 2024); and attacks such as DiffPrivate are computationally heavy, requiring significant run time to attain high success rates (see §6). Due to these limitations, existing attacks may fail to uncover weakness in FR or may not lend themselves to being incorporated in training schemes for improving FR's robustness.

For specific forms of adversarial examples, such as those created by adversarial perturbations with bounded $\ell_p$-norms, numerous defense types like adversarial training (Wong et al., 2020) and randomized smoothing (Cohen et al., 2019) can help boost robustness. However, to our knowledge, no established defenses have been proposed to mitigate general semantic attacks (Le & Carlsson, 2025). Particularly, adversarial training for defending against general semantic attacks remains infeasible due to the run-time overhead or limited success rates of existing attacks. Consequently, FR remains vulnerable to general semantic attacks.

To fill these gaps, this work presents a new general semantic attack, BOUNDSTYLE, and a defense, STYLEAT. Our attack takes advantage of StyleGAN3's rich latent space, among others, to produce high-fidelity semantic edits to fool FR. Importantly, BOUNDSTYLE is tunable, enabling us to control the magnitude of edits and run time, thus ensuring that identity is preserved w.r.t. human observer and allowing us to execute time-efficient variants. Notably, we also find that BOUNDSTYLE is highly successful, on par with the state-of-the-art DiffPrivate attack (Le & Carlsson, 2025), while being significantly ($\approx \times 9.5$) faster. Interestingly, while we find compelling evidence that DiffPrivate introduces imperceptible adversarial perturbations alongside visible semantic edits, we find no clear signs that BOUNDSTYLE makes edits other than semantic ones. Altogether, BOUNDSTYLE's advantages render it suitable for measuring the susceptibility of FR models to attacks as well as for adversarial training to help improve robustness against semantic attacks.

Our defense, STYLEAT, employs a time-efficient variant of BOUNDSTYLE to adversarially train FR models and improve their adversarial robustness against general semantic attacks. Against BOUND-STYLE, STYLEAT achieves up to 28.6% increase in robust accuracy (depending on the setting explored) compared to undefended models, markedly higher than defenses not tailored for general semantic attacks that achieve $\leq 6.0\%$ increase in robust accuracy. Crucially, STYLEAT also leads to improvements against DiffPrivate, an attack not encountered during training, with up to 46.3% higher robust accuracy than undefended models, showcasing that STYLEAT generalizes to unknown general semantic attacks.

We next present related work (§2) and lay out our threat model (§3). Subsequently, we present the technical approach behind BOUNDSTYLE and STYLEAT (§4) before presenting our experimental results (§5–6) and concluding (§7).

## 2 RELATED WORK

**Attacks on FR** Prior work has demonstrated that ML models in general, and FR in particular, are vulnerable to test-time evasion attacks that induce misclassifications via imperceptible adversarial perturbations with bounded $\ell_p$-norm (Szegedy et al., 2014). For instance, the fast gradient sign method (FGSM) creates attacks by perturbing inputs in the gradient direction once (Goodfellow et al., 2015), while projected gradient descent (PGD) does so through multiple, iterative perturbations Madry et al. (2018). However, such attacks may be challenging to realize in real-world settings due to difficulties in implementing norm-bounded noise and cameras' sampling errors, among others (Sharif et al., 2016). To this end, researchers have proposed semantic attacks—attacks that alter inputs in minor, easy-to-realize, and semantically meaningful ways—to mislead FR models.

Semantic attacks consist of two families. The first family of attacks makes *ad hoc* changes to inputs, for example, by introducing adversarial accessories like eyeglasses or hats to fool models (Komkov & Petiushko, 2021; Sharif et al., 2016). These also include attacks that fool models via facial make-up or spatial transformations applied in an adversarial manner (Hu et al., 2022; Xiao et al., 2018; Yin et al., 2021). By contrast, the second family of attacks leverages *general* edits of inputs to induce misclassifications, including, but not limited to, changes of expression, age, and accessories, or a combination thereof (Barattin et al., 2023; Jia et al., 2022; Le & Carlsson, 2024; 2025; Liu et al., 2024). Our work focuses on general semantic attacks, proposing a new attack and a defense.

General semantic attacks typically leverage generative models to produce adversarial edits of inputs. For instance, attacks such as StyleAdv (Le & Carlsson, 2024) and Adv-Attribute (Jia et al., 2022) search for adversarial editing directions in the latent space of generative adversarial networks (GANs) to produce misclassifications. By contrast, Adv-Diffusion (Liu et al., 2024) and DiffPrivate (Le & Carlsson, 2025) use latent diffusion models to find adversarial semantic edits of inputs. DiffPrivate is the most recent and potent general semantic attack; we use it in our evaluation.

**Defending FR** A diversity of defenses against evasion attacks have been proposed, including, but not limited to, ones that detect attacks (e.g., Metzen et al. (2017)); filter out adversarial perturbations (e.g., Xu et al. (2018)); smoothen classification boundaries to reduce model vulnerability (e.g., Carlini et al. (2023); Cohen et al. (2019)); verify robustness against specific adversaries (e.g., Katz et al. (2019)); and adversarial train of models by injecting correctly labeled adversarial inputs to the training data to inherently increase model robustness (e.g., Kurakin et al. (2017); Madry et al. (2018)). Due to its intuitive nature, its ability to improve adversarial robustness in a practical man-

ner against different attack types, and absence of impact on model's inference time, Adversarial training is particularly appealing and was widely studied. Still, adversarial training may be computationally expensive due to the overhead of producing attacks during training, potentially rendering training prohibitive. To this end, researchers have also explored efficient adversarial training variants (e.g., Shafahi et al. (2019); Wong et al. (2020)). We take inspiration from Wong et al. (2020) who showed how to leverage the efficient FGSM attack in training to induce robustness against much more potent attacks at test time.

To the best of our knowledge, there are *no established defenses for countering general semantic attacks against FR*. Nonetheless, several countermeasures have been proposed to counter ad hoc semantic attacks. For example, defense through occlusion attack (DOA) adversarially trains models with carefully positioned patches containing adversarial patterns to help counter eyeglass attacks (Wu et al., 2020a). As another example, DiffPure utilizes forward diffusion followed by image recovery to remove adversarial manipulations, helping counter spatial adversarial modifications (Nie et al., 2022). Prior work has also shown that input filters, such as JPEG compression and blurring can hinder general semantic attacks to some degree when those attacks are agnostic to the filters (Le & Carlsson, 2025). Related to our work, Laidlaw et al. (2020) adversarially train ML models with imperceptible adversarial perturbations created using generative models. However, they only evaluate robustness against imperceptible perturbations and spatial manipulations.

## 3 THREAT MODEL

We consider an adversary carrying out a general semantic attack against FR. Per standard practice, we assume the FR system is tuned to an operating point where the false positive rate (FPR) is below a target threshold, such as 0.01 FPR (Introna & Nissenbaum, 2009; Le & Carlsson, 2025). We consider a powerful adversary aiming to produce arbitrary, untargeted misclassifications (primarily, false negatives) rather than impersonations (i.e., targeted attacks), as, intuitively, this adversary would be more challenging to defend against. Contrastively, our defender aims to hinder the adversary's attempts through keeping the robust true positive rate (TPR)—i.e., the TPR under attacks—high while preserving the benign accuracy of standard FR models when ingesting clean inputs. In line with Le & Carlsson (2025), We primarily focus on potent adversaries with white-box access to both the FR model and the defense, but we also consider black-box adversaries without access to the model or defense that seek to transfer attacks from surrogate models, as well as gray-box adversaries that have access to the (undefended) model but not to the defense.

## 4 TECHNICAL APPROACH

We now present our proposed attack (BOUNDSTYLE) and defense (STYLEAT).

### 4.1 BOUNDSTYLE: A HIGH-FIDELITY, POTENT, TUNABLE GENERAL SEMANTIC ATTACK

We design BOUNDSTYLE as a general semantic attack against FR based on generative models with three goals in mind: *(1)* We require that the attack produces evasive face images with *high visual fidelity* through diverse and realistic edits; *(2)* We seek *tunability* such that we would be able to control the run time of the attack to later enable efficient adversarial training and bound the magnitude of the edit so as the identity in the face image is unchanged (for a human observer); and *(3)* We need the attack to be *potent* exposing the weaknesses of FR through achieving high success rates. We next describe how our design of BOUNDSTYLE ensures high visual fidelity and tunability. Our experiments (§5–6) evidence the attack's potency.

**A Tunable Attack** Let $F$ denote the feature extractor used for FR, $C$ the preprocessing algorithm (cropping and alignment), $G$ a generative model, $x$ the image to modify with an inverted latent code $l$, and $x_e$ a face image of the same subject enrolled in the gallery. BOUNDSTYLE aims to edit $x$ through a slight modification of $l$ such that the similarity (sim, usually cosine similarity) with $x_e$ would become small. Formally, BOUNDSTYLE aims to minimize the following loss through a perturbation $\delta$ of the latent code:

$$L_{atk} = \text{sim}(F(C(G(l + \delta), F(C(x_e))).$$

BOUNDSTYLE optimizes the loss through iterative gradient-based optimization, in the spirit of PGD, and its performance is governed by two primary inputs $T$, the number of iterations, and $\beta$, the magnitude (specifically, $\ell_2$-norm) of the perturbation $\delta$. Initially, $\delta_0$ is randomly initialized inside the $\beta$-ball, as random initialization is critical to the performance of evasion attacks in adversarial training (Wong et al., 2020). Subsequently, in each iteration (up to $T$), BOUNDSTYLE updates $\delta_i = \delta_i - \alpha \cdot \frac{g}{\|g\|_\infty}$ where $g = \nabla_{\delta_i} L_{atk}$ is the loss gradient and $\alpha$ is a step-size hyperparameter. At any point, if the norm of $\delta_i$ exceeds the bound $\beta$, it is projected back to the $\beta$-ball via $\delta_i = \beta \cdot \frac{\delta_i}{\|\delta_i\|_2}$.

Both $T$ and $\beta$ are tunable parameters that enable achieving different trade-offs with BOUNDSTYLE. Decreasing $T$ may potentially harm the attack success, but also makes BOUNDSTYLE faster, rendering it more suitable for (efficient) adversarial training. Moreover, setting $\beta$ should balance two goals—it should be large enough so that the attack is successful due to stronger edits in the latent space, but small enough so that the identity of the subject is preserved w.r.t. human observers.

**Ensuring High Fidelity** We take several measures to ascertain that BOUNDSTYLE introduces high-fidelity edits. First, we adopt the StyleGAN3 generator (Karras et al., 2021), which provides a rich latent space with diverse editing directions and high quality outputs. Second, we invert $x$ to a latent code $l$ that maps back almost precisely to $x$, thus preserving identity. To do so, we use a hybrid combination of encoder-based projection to the latent space (Alaluf et al., 2022) followed by direct gradient-based optimization for accurate reconstruction of the face image (Zhu et al., 2020). Last, we use pivotal tuning, a method that tunes the generator $G$ to enable better editability while preserving identity (Roich et al., 2022).

## 4.2 STYLEAT: STYLE-AWARE ADVERSARIAL TRAINING

Building on BOUNDSTYLE, we design STYLEAT, a method for adversarially training FR models to enhance their adversarial robustness against general semantic attacks. In particular, STYLEAT fine-tunes pre-trained FR feature extractors while aiming to balance three different objectives: *(1)* Preserving benign accuracy on clean images; *(2)* Improving robust accuracy against general semantic attacks; and *(3)* Countering imperceptible perturbations inadvertently introduced by certain established semantic attacks. To achieve each of these goals, STYLEAT minimizes the triplet losses $L_{Cln}$, $L_{AdvSem}$, and $L_{AdvPix}$, respectively. These losses are balanced through non-negative hyperparameters that accumulate to one (i.e., $\lambda_{Cln} + \lambda_{AdvSem} + \lambda_{AdvPix} = 1$). We next elaborate how each of $L_{AdvSem}$ and $L_{AdvPix}$ are computed and optimized and how we select triplets for loss computation; minimizing $L_{Cln}$ is intuitive and follows standard practice (Wang et al., 2021). An overview of STYLEAT's pipeline is depicted in Fig. 1; Alg. 1 in App. A presents its pseudocode.

**Computing and Optimizing $L_{AdvSem}$** We leverage BOUNDSTYLE to produce general semantic attacks for adversarial training. However, as executing the most potent attack variant during training may make the training process infeasible, we incorporate "weakened" but efficient attack variants into training. Specifically, we run fast variants of BOUNDSTYLE with a small number of iterations $T$, analogously to fast adversarial training with FGSM (Wong et al., 2020). Here, we tune $T$ and the step size $\alpha$ such that training can be completed within a few days under our resource constraints, while the attacks are sufficiently evasive to help enhance FR's robustness against general semantic attacks.

**Computing and Optimizing $L_{AdvPix}$** Our evaluation of established semantic attacks shows that certain attacks may introduce imperceptible perturbations alongside semantic edits to mislead FR. This phenomenon is perhaps most clearly demonstrated when evaluating attacks against filter-based defenses such as JPEG compression that primarily affect imperceptible, non-semantic perturbations. Said differently, if such filters have a pronounced effect on an attack's success, one may conclude that misclassification did not occur due to a semantic edit, but rather due to imperceptible changes of pixels. Indeed, our evaluation shows that DiffPrivate exhibits a significant decrease in their success once JPEG compression and similar filters are employed (see Fig. 4 and Fig. 10, with App. C providing additional support through a frequency-domain analysis). To account for these potential perturbations, we also adversarially train our models against imperceptible $\ell_\infty$-norm-bounded adversarial perturbations. We find that doing so does not harm robustness against semantic attacks that do not seem to introduce imperceptible adversarial perturbations (namely, BOUNDSTYLE; App. C).

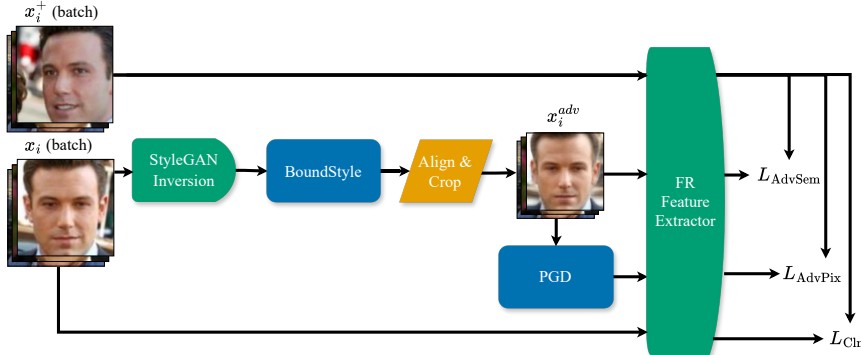

Figure 1: **An overview of STYLEAT's training pipeline.** The framework processes a mini-batch of identity pairs $\{(x_i, x_i^+)\}$ through three parallel branches: (1) The *clean branch* extracts features from the original samples to preserve benign accuracy ($L_{\text{Cln}}$). (2) The *latent branch* generates a batch of semantic adversarial examples $x_i^{\text{adv}}$ via BOUNDSTYLE to improve robustness against semantic attacks ($L_{\text{AdvSem}}$). (3) The *pixel branch* applies imperceptible perturbations to the semantically edited batch to increase robustness against imperceptible perturbations ($L_{\text{AdvPix}}$). Batch processing enables hard negative mining to optimize the triplet losses.

Toward countering imperceptible adversarial perturbations, we integrate fast PGD attacks with few iterations into training, in the spirit of Wong et al.'s (2020) work. Importantly, to avoid robust overfitting, we perform PGD with random initialization before each attack. Crucially, we apply PGD to images already edited adversarially with BOUNDSTYLE, as we aim to counter the combination of adversarial semantic edits and imperceptible perturbations.

**Selection of Triplets** For a given positive pair of samples depicting the same identity, we select the hardest negative sample from the batch to compute triplet losses. Doing so, as shown in prior work on adversarially robust metric learning (Mao et al., 2019), is most conducive for adversarial robustness. More precisely, to compute the triplet loss, for a positive pair of samples $p$ and $a$ (standing for positive and anchor, respectively), we select the negative sample $n$ from the batch such that it depicts a different identity and is most similar to $a$ in the feature space compared to other samples in the batch. Subsequently, the triplet loss is calculated by $(\text{sim}(F(C(n)), F(C(a))) - \text{sim}(F(C(p)), F(C(a))) + \mu)^+$, where $\mu$ is a small (non-negative) constant. Moreover, in the interest of improving adversarial robustness, we adversarially perturb the anchor sample $a$ when computing $L_{\text{AdvSem}}$ and $L_{\text{AdvPix}}$, and select the hardest negative sample after applying the perturbations, per Li et al. (2019).

## 5 EXPERIMENTAL SETUP

**FR Backbones** We employ seven popular and high-performing FR models, four convolutional networks, one vision transformer, and two convolutional networks equipped with specialized supervisory heads. Specifically, we use MobileFaceNet (MobileFace) (Chen et al., 2018); ResNet (ResNet-152, IR-SE) (He et al., 2016); RepVGG (Ding et al., 2021); LightCNN (Wu et al., 2018; 2020b); Swin Transformer (SwinT) (Liu et al., 2021); and ArcFace (Deng et al., 2019) as well as MagFace (Meng et al., 2021) with MobileFace backbones. We use these models in two roles, both as targets for attacks, and as surrogates (i.e., proxies) for producing transferable adversarial examples. As part of STYLEAT, we create adversarially trained variants of ResNet and RepVGG through fine-tuning the original pre-trained backbones. We obtain the initial weights from FaceX-Zoo (Wang et al., 2021).

**Datasets** Following FaceX-Zoo (Wang et al., 2021), we construct our training dataset from MS-Celeb-1M-v1c (Guo et al., 2016), we use their randomly selected preprocessed positive image pairs for training. We then run our preprocessing pipeline on all images and discard pairs where an image fails face or landmark detection, yielding a final training set of 52,269 positive image pairs. (Note

that negative images are selected as hard negatives, independently for each batch during training.) We evaluate on two widely used datasets: *(1)* Labeled Faces in the Wild (LFW) (Huang et al., 2007) and *(2)* VGG-Face (Parkhi et al., 2015). Specifically, we select 216 and 156 positive pairs from LFW and VGG-Face, respectively. When selecting these images, we ensure no overlap between the selected identities and those appearing in the training (and pre-training) sets from MS-Celeb-1M-v1c, by removing any image that has similarity with any training samples above the threshold where the original ResNet backbone has an FPR of $10^{-4}$.

**Attacks** We evaluate BOUNDSTYLE, bounding the edit perturbation $\ell_2$-norms in StyleGAN3's latent space to $\beta \in \{1.0, 1.5, 2.0, 3.0\}$. We avoid perturbations of larger magnitude to help preserve subject identities in images (App. B). For best performance, we set the step size $\alpha=\beta$, and the number of iterations $T$=30, as more iterations show no improvements in attack success (App. C). As a baseline, we evaluate DiffPrivate (Le & Carlsson, 2025), a state-of-the-art attack that edits images through perturbations in the diffusion latent $z$-space. For a fair comparison with BOUNDSTYLE, we adopt a norm-bounded variant of DiffPrivate by enforcing $\|\Delta z\|_2 = \|z_{\mathrm{adv}} - z_0\|_2 \in \{1, 2, 3, 4, 5, 6\}$. We avoid perturbations with norm $>6$ to preserve subject identities in images (App. B). Under this bounded setup, we find that capping the optimization at 70 iterations for convergence to the highest attack success (App. C).

**Defenses** We apply STYLEAT on both ResNet and RepVGG models, adversarially training them from checkpoints obtained from FaceX-Zoo. We use a low-cost variant of BOUNDSTYLE for adversarial training, with $T = 3$ attack iterations, $\beta = 3$ $\ell_2$-norm for perturbations in the StyleGAN3 latent space, and $\alpha = 3$ as step size in the attack; we run PGD attacks for 2 iterations with $\epsilon = 24/255$ $\ell_\infty$-norm for perturbations in the pixel space and step size $\alpha = \epsilon/2$. We find that these parameters help attain reasonable benign accuracy within feasible time under our resource constraints. After hyperparameter search, we set specific loss weights for each backbone: for ResNet, we set $\lambda_{\mathrm{Cln}} = 0.35$, $\lambda_{\mathrm{AdvSem}} = 0.45$, and $\lambda_{\mathrm{AdvPix}} = 0.2$; for RepVGG, we set $\lambda_{\mathrm{Cln}} = 0.1$, $\lambda_{\mathrm{AdvSem}} = 0.8$, and $\lambda_{\mathrm{AdvPix}} = 0.1$. Other parameters (e.g., for the optimizer and triplet loss margin) are adopted from FaceX-Zoo. We run training on 8 NVIDIA RTX A5000 GPUs with a batch size of 4 per GPU (global batch size 32). We run training for 8 epochs, completing it within 4 days. As baselines, we compare STYLEAT with DOA—a defense tailored for ad hoc semantic attacks using eyeglasses (see §2)—and standard filters considered in prior work (Le & Carlsson, 2025), including Gaussian blur, denoising via total variation minimization, JPEG compression, feature squeezing, spatial smoothing, and random noise injection.

**Metrics and FR Operating Point** We evaluate model robustness through accuracy (i.e., TPR) after perturbing one of the samples from a positive pair. Following standard practice, we calibrate the verification threshold to meet a target FPR on clean data (without attacks). Specifically, we perform the calibration on LFW's full (clean) validation set (containing negative and positive pairs) to obtain an FPR of 0.01, similar to Le & Carlsson (2025).

# 6 EXPERIMENTAL RESULTS

We now evaluate the BOUNDSTYLE attack and STYLEAT defense.

## 6.1 BOUNDSTYLE IS POTENT AND FAST

**White-box Attacks** Tab. 1 reports the robust accuracy of the seven (undefended) FR backbones against BOUNDSTYLE and DiffPrivate on the LFW and VGG-Face datasets when varying the attack budgets. The results show that both attacks are potent in the white-box setting—despite high benign accuracy ($>99\%$), the robust accuracy drops as the attack budgets increase. At their strongest BOUNDSTYLE budgets ($\beta$=3 for BOUNDSTYLE and $\|\Delta z\|$=6 for DiffPrivate), the average robust accuracy against BOUNDSTYLE ($\sim$46% on LFW and $\sim$37% on VGG-Face) is slightly higher than against DiffPrivate ($\sim$35.5% on LFW and $\sim$16% on VGG-Face).

**Black-box Attacks** Fig. 2 presents the robust accuracy of models when transferring attacks between models at the highest attack budgets considered. In the heatmaps, off-diagonals show robust accuracy in black-box settings (transfer from the rows to columns), while diagonals correspond to white-box settings. BOUNDSTYLE *exhibits strong transferability*—on both datasets, the heatmaps

**LFW**

| Model | Clean | BoundStyle — $\beta$ | | | | DiffPrivate — $\|\Delta z\|$ | | | | | |
|---|---|---|---|---|---|---|---|---|---|---|---|
| | | 1 | 1.5 | 2 | 3 | 1 | 2 | 3 | 4 | 5 | 6 |
| SwinT | 99.1 | 94.9 | 88.0 | 78.7 | 52.6 | 99.5 | 98.6 | 98.6 | 85.6 | 50.5 | 49.1 |
| LightCNN | 99.1 | 90.3 | 84.7 | 72.7 | 40.5 | 99.1 | 97.7 | 89.4 | 62.0 | 32.9 | 28.7 |
| MobileFace | 99.1 | 91.7 | 88.0 | 75.0 | 50.2 | 98.6 | 98.1 | 96.3 | 79.2 | 53.7 | 38.0 |
| RepVGG | 99.1 | 92.6 | 83.3 | 69.4 | 40.5 | 99.1 | 99.1 | 96.3 | 72.7 | 43.5 | 38.0 |
| ResNet | 99.1 | 93.5 | 83.8 | 70.8 | 39.5 | 99.1 | 98.6 | 93.5 | 71.8 | 41.7 | 33.8 |
| MagFace | 99.1 | 94.0 | 88.9 | 77.8 | 52.2 | 98.6 | 98.6 | 96.3 | 74.5 | 44.9 | 28.7 |
| ArcFace | 99.5 | 94.4 | 85.2 | 74.1 | 45.0 | 99.1 | 98.6 | 91.7 | 69.0 | 38.4 | 31.5 |

**VGG-Face**

| Model | Clean | BoundStyle — $\beta$ | | | | DiffPrivate — $\|\Delta z\|$ | | | | | |
|---|---|---|---|---|---|---|---|---|---|---|---|
| | | 1 | 1.5 | 2 | 3 | 1 | 2 | 3 | 4 | 5 | 6 |
| SwinT | 100.0 | 85.3 | 80.8 | 65.4 | 45.5 | 83.9 | 76.8 | 64.5 | 41.9 | 26.5 | 19.4 |
| LightCNN | 98.7 | 84.0 | 72.4 | 60.3 | 34.0 | 69.7 | 58.7 | 45.8 | 25.2 | 18.1 | 14.8 |
| MobileFace | 97.4 | 78.8 | 72.4 | 62.2 | 38.5 | 70.3 | 66.5 | 47.7 | 34.2 | 21.9 | 16.1 |
| RepVGG | 100.0 | 80.8 | 71.2 | 60.3 | 32.0 | 77.4 | 69.7 | 53.5 | 32.3 | 22.6 | 17.4 |
| ResNet | 100.0 | 84.0 | 72.4 | 59.6 | 34.6 | 77.4 | 67.7 | 51.0 | 30.3 | 22.6 | 17.4 |
| MagFace | 98.1 | 75.6 | 67.3 | 57.7 | 40.8 | 69.0 | 67.7 | 52.3 | 31.0 | 20.0 | 14.8 |
| ArcFace | 98.7 | 80.8 | 71.8 | 52.6 | 34.4 | 71.6 | 58.7 | 44.5 | 27.7 | 20.0 | 14.2 |

Table 1: **Robust accuracy of models against white-box attacks** (higher is better). We report results on LFW and VGG-Face while varying the attack budgets of BoundStyle and DiffPrivate. The second column from the left reports the benign accuracy (without attacks).

are near-uniform with off-diagonals typically within 5–15% of the diagonal. This indicates strong transferability, where attacks crafted on one model substantially reduce accuracy on others. By contrast, DiffPrivate exhibits relatively weak transferability—off-diagonals are often 20–60% higher than the diagonal, meaning attacks fail to carry over. For instance, on LFW, evaluating on SwinT yields 49% robust accuracy in the white-box setting compared to 90–97% robust accuracy when transferring the attack from the convolutional networks to SwinT, suggesting that DiffPrivate is architecture-specific.

**Attacks' Run Times** We benchmark attacks' run times on an NVIDIA RTX A6000 GPU, executing each attack 100 times and averaging the run time. For fair comparison, we use an equal batch size of 1 for both attacks. Under this setting, BoundStyle takes an average of 8,302 ms to complete per image compared to an average of 79,094 ms attained by DiffPrivate, demonstrating approximately $\times 9.5$ speedup. This result highlights BoundStyle's better fit for adversarial training compared to other state-of-the-art attacks: BoundStyle is not only roughly as powerful in inducing misclassifications in white-box settings and more transferable in black-box settings, but it is also significantly faster. We further analyze the trade-offs between wall-clock time and attack success in App. C, finding that BoundStyle is effective even under strict time constraints where DiffPrivate fails.

## 6.2 STYLEAT IMPROVES ADVERSARIAL ROBUSTNESS

**White-box Setting** We execute white-box attacks against STYLEAT and DOA as well as against the undefended models (ResNet and RepVGG) at different attack budgets. Tab. 2 reports the results. Compared to the undefended models, STYLEAT shows a 0.4–20.9% increase in robust accuracy on the different attack budgets against both BoundStyle and DiffPrivate, with the latter unseen during training. The DOA defense, tailored for ad hoc semantic attacks, shows a 0.4–0.7% higher robust accuracy than STYLEAT a few attack budgets on DiffPrivate, but otherwise trails behind STYLEAT's robust accuracy by 0.4–9.6% against the DiffPrivate attack. However, against BoundStyle, DOA is counterproductive for most attack budgets, decreasing robust accuracy compared to the undefended model by up to 6%. Altogether, these results show that STYLEAT reliably improves

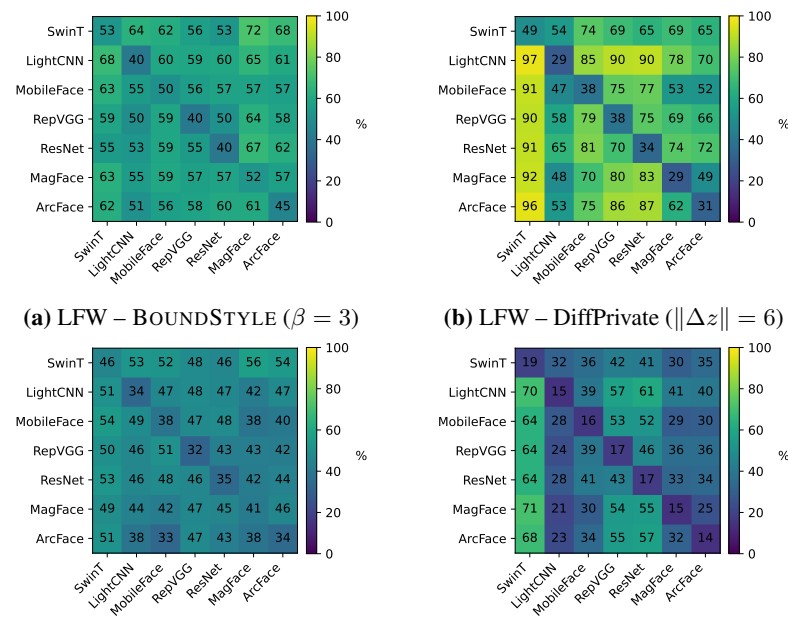

**(a)** LFW – BOUNDSTYLE ($\beta = 3$)      **(b)** LFW – DiffPrivate ($\|\Delta z\| = 6$)

**(c)** VGG-Face – BOUNDSTYLE ($\beta = 3$)    **(d)** VGG-Face – DiffPrivate ($\|\Delta z\| = 6$)

Figure 2: **Robust accuracy of models against black-box attacks**. Each heatmap reports the robust accuracy of the model at the column, when transferring attacks from the model at the row. Diagonals correspond to white-box attacks.

**LFW**

| Backbone | Defense | Clean | BOUNDSTYLE — budget $\beta$ | | | | DiffPrivate — $\|\Delta z\|$ | | | | | |
|---|---|---|---|---|---|---|---|---|---|---|---|---|
| | | | 1 | 1.5 | 2 | 3 | 1 | 2 | 3 | 4 | 5 | 6 |
| ResNet | No Defense | 99.1 | 93.5 | 83.8 | 70.8 | 39.5 | 99.1 | 98.6 | 93.5 | 71.8 | 41.7 | 33.8 |
| | DOA | 99.1 | 94.9 | 80.6 | 69.4 | 35.4 | 99.1 | 98.6 | 96.8 | 85.6 | **54.6** | 42.6 |
| | STYLEAT | **99.5** | **97.7** | **91.7** | **84.3** | **50.7** | **99.5** | **99.5** | **97.7** | **86.6** | 54.2 | **47.2** |
| RepVGG | No Defense | 99.1 | 92.6 | 83.3 | 69.4 | 40.5 | 99.1 | 99.1 | 96.3 | 72.7 | 43.5 | 38.0 |
| | DOA | **99.5** | 93.1 | 84.3 | 72.7 | 34.5 | **99.5** | 99.1 | 97.7 | 89.4 | 59.7 | 43.1 |
| | STYLEAT | **99.5** | **97.2** | **92.1** | **84.7** | **54.3** | **99.5** | **99.5** | **98.6** | **92.6** | **64.4** | **50.0** |

**VGG-Face**

| Backbone | Defense | Clean | BOUNDSTYLE — budget $\beta$ | | | | DiffPrivate — $\|\Delta z\|$ | | | | | |
|---|---|---|---|---|---|---|---|---|---|---|---|---|
| | | | 1 | 1.5 | 2 | 3 | 1 | 2 | 3 | 4 | 5 | 6 |
| ResNet | No Defense | **100.0** | 84.0 | 72.4 | 59.6 | 34.6 | 77.4 | 67.7 | 51.0 | 30.3 | 22.6 | 17.4 |
| | DOA | 99.4 | 82.7 | 69.9 | 55.8 | 28.9 | 79.4 | 74.2 | 58.7 | 38.1 | 27.1 | 16.1 |
| | STYLEAT | **100.0** | **88.5** | **81.4** | **65.4** | **35.3** | **81.3** | **76.1** | **63.2** | **41.9** | **27.7** | **18.1** |
| RepVGG | No Defense | **100.0** | 80.8 | 71.2 | 60.3 | 32.0 | 77.4 | 69.7 | 53.5 | 32.3 | 22.6 | 17.4 |
| | DOA | 98.7 | 79.5 | 73.7 | 57.0 | 27.6 | **81.3** | 74.2 | 57.4 | 40.0 | 26.5 | **21.3** |
| | STYLEAT | 99.4 | **89.1** | **80.8** | **69.9** | **35.7** | 80.6 | **76.1** | **62.6** | **43.2** | **31.0** | 20.6 |

Table 2: **Evaluating defenses against white-box attacks**. For each defense and undefended model, we report benign accuracy and robust accuracy under varied attack budgets on the LFW and VGG-Face datasets, for both ResNet and RepVGG backbones.

robust accuracy against general semantic attacks while generalizing to attacks unseen during training. Importantly, STYLEAT also maintains the benign accuracy on clean data as the undefended model or even roughly improves it.

**Gray-box Setting** We also evaluate STYLEAT and filter-based defenses against gray-box attacks, where the adversary produces attacks against the undefended models (ResNet and RepVGG), in a manner agnostic to the defense. Figs. 3–4 report the robust accuracy achieved against BOUND-STYLE and DiffPrivate, respectively, with the ResNet backbone. Against BOUNDSTYLE, it can be seen that filter-based defenses have little impact on robustness, increasing robust accuracy by 6.0% in the best case compared to the undefended model. In comparison, STYLEAT results in up to 33.3% increase in robust accuracy. The filter-based defenses are more useful against DiffPrivate, increasing robust accuracy by up to 31.5%. Nonetheless, STYLEAT is also more effective than filter-based defenses against DiffPrivate, with an increase of up to 50.9% in robust accuracy compared to the undefended model. These results further highlight STYLEAT's utility against defense-agnostic adversaries including against attacks not accounted for during training (i.e., DiffPrivate). The results on the RepVGG backbone (Figs. 9–10 in App. C) show identical trends.

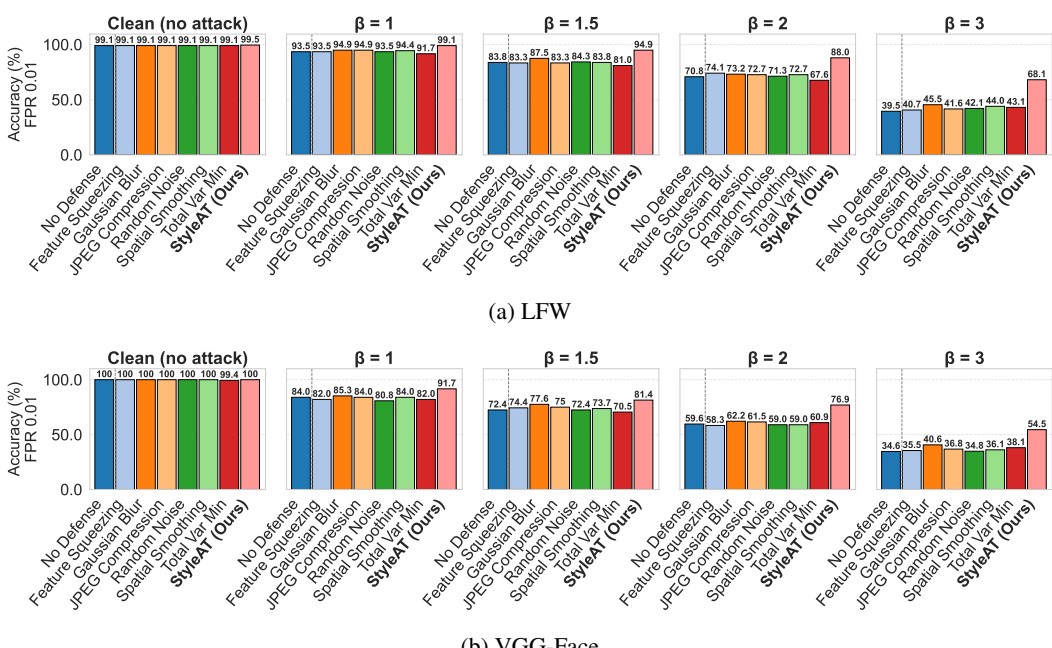

Figure 3: **Evaluating defenses against gray-box BOUNDSTYLE attacks (ResNet).**

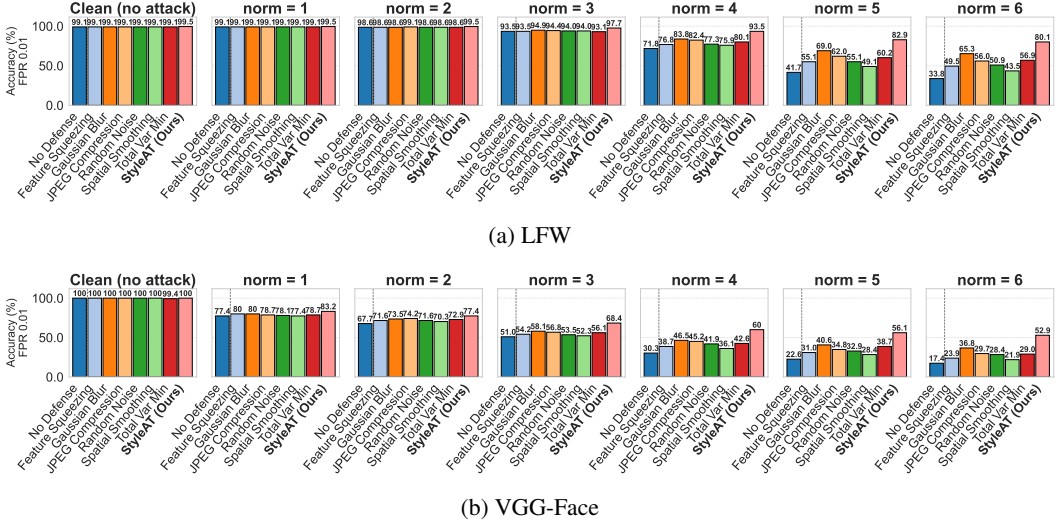

Figure 4: **Evaluating defenses against gray-box DiffPrivate attacks (ResNet).**

## 7    CONCLUSION, LIMITATIONS, AND FUTURE WORK

Our work studies general semantic attacks against FR, proposing a new attack and a defense. The proposed attack, BOUNDSTYLE, produces high-fidelity images, attains high success rates, and is tunable, lending itself to being integrated into adversarial training. We find that BOUNDSTYLE achieves success rates on par with state-of-the-art attacks and even outperforms them in some settings, while being almost $\times 9.5$ more time-efficient. Our defense, STYLEAT, is, to the best of our knowledge, the first defense tailored for general semantic attacks against FR. STYLEAT builds on BOUNDSTYLE, augmenting the training data with evasive samples produced via a weakened but efficient attack variant, leading to significant increases in robust accuracy in several settings we consider.

While STYLEAT helps improve adversarial robustness of FR, thus improving security, it is important to notice that it does not prevent attacks completely. Moreover, in line with other adversarial training methods, STYLEAT is an empirical and practical defense, but it does not provide provable security guarantees. Future work may seek to further increase adversarial robustness against general semantic attacks and derive theoretical guarantees, for instance through randomized smoothing (Cohen et al., 2019) in the latent space.

### ETHICS STATEMENT

Our work lies in the general field of ML security. The two main innovations we make, BOUND-STYLE and STYLEAT, can be used to challenge the security of FR and improve it, respectively. While BOUNDSTYLE may be used to mislead existing deployment of FR, we argue that developing it brings about more advantages than disadvantages. In particular, if not created by us, adversaries may use methods similar to BOUNDSTYLE to fool FR systems without our knowledge. Crucially, BOUNDSTYLE is a critical component in enabling STYLEAT, allowing us to enhance FR's reliability in the face of general semantic attacks.

### REPRODUCIBILITY STATEMENT

As proponents of open science, we intend to release our code, data, and models upon paper acceptance to facilitate reproducibility and allow future work to build off of our contributions. Our code release will be accompanied with detailed documentation and scripts that can be used to recreate all our results. We also hope that our implementation would help practitioners improve FR's reliability in practice.

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

## A  STYLEAT'S ALGORITHM

Alg. 1 presents the pseudocode of STYLEAT.

---

**Algorithm 1** STYLEAT (per minibatch)

---

**Input:** minibatch $\{(x_i, x_i^+)\}_{i=1}^B$; StyleGAN encoder $E$ and generator $G$, feature extractor $f_\theta$, latent attack steps $K$, step size $\alpha$, attack strength $\beta$, triplet loss margin $\mu$, pixel PGD steps $S$, $\alpha_{\text{pix}}$ step, PGD radius $\varepsilon$;

**loss weights:** $\lambda_{\text{Cln}}, \lambda_{\text{AdvSem}}, \lambda_{\text{AdvPix}} \geq 0$ with $\lambda_{\text{Cln}} + \lambda_{\text{AdvSem}} + \lambda_{\text{AdvPix}} = 1$

**Output:** updated parameters $\theta$

---

1: **for** $i = 1$ **to** $B$ **do**
2:     $l_i \leftarrow \text{OptimizeInv}\big(E(x_i), G\big)$                                               {inversion (offline cacheable)}
3: **end for**
4: **for** $i = 1$ **to** $B$ **do**
5:     **Latent branch (BOUNDSTYLE):**
6:     Sample unit direction $v_i \sim \mathcal{N}(0, I)$; $v_i \leftarrow v_i / \|v_i\|_2$
7:     Sample $b_i \sim \mathcal{U}(0, \beta)$;
8:     $\delta_i \leftarrow b_i v_i$
9:     $x_i \leftarrow \text{CropAlign}(x_i)$;     $x_i^+ \leftarrow \text{CropAlign}(x_i^+)$
10:     **for** $k = 1$ **to** $K$ **do**
11:        $x_i^{\text{adv}} \leftarrow \text{CropAlign}\big(G(l_i + \delta_i)\big)$
12:        $g \leftarrow \nabla_\delta \big[-\cos\big(f_\theta(x_i^{\text{adv}}), f_\theta(x_i)\big)\big]$                               {dodging: reduce cosine}
13:        $\delta_i \leftarrow \big(\delta_i + \alpha \frac{g}{\|g\|_\infty}\big)$
14:        Project $\delta_i \leftarrow \beta \cdot \frac{\delta_i}{\|\delta_i\|_2}$
15:     **end for**
16:     $x_i^{\text{adv}} \leftarrow \text{CropAlign}\big(G(l_i + \delta_i)\big)$
17:     **Pixel branch (PGD under $\ell_\infty(\varepsilon)$, *around* $x_i^{\text{adv}}$):**
18:     $\hat{x}_i^{(0)} \leftarrow \text{Clip}\big(x_i^{\text{adv}} + \mathcal{U}[-\varepsilon, \varepsilon]\big)$                                 {random init near $x_i^{\text{adv}}$}
19:     **for** $s = 1$ **to** $S$ **do**
20:        $g_x \leftarrow \nabla_x \Big[\cos\big(f_\theta(\hat{x}_i^{(s-1)}), f_\theta(x_i^+)\big)\Big]$
21:        $\hat{x}_i^{(s)} \leftarrow \hat{x}_i^{(s-1)} - \alpha_{\text{pix}} \cdot \text{sign}(g_x)$
22:        $\hat{x}_i^{(s)} \leftarrow \Pi_{B_\infty(x_i^{\text{adv}}, \varepsilon)}\big(\hat{x}_i^{(s)}\big)$                            {project to $\ell_\infty$ ball *around* $x_i^{\text{adv}}$}
23:        $\hat{x}_i^{(s)} \leftarrow \text{Clip}\big(\hat{x}_i^{(s)}\big)$                                      {valid pixel range}
24:     **end for**
25:     $x_i^{\text{pix}} \leftarrow \hat{x}_i^{(S)}$
26: **end for**
27: **for** $i = 1$ **to** $B$ **do**
28:     $e_i^+ = f_\theta(x_i^+)$; $e_i^{\text{cln}} = f_\theta(x_i)$; $e_i^{\text{advSem}} = f_\theta(x_i^{\text{adv}})$; $e_i^{\text{advPix}} = f_\theta(x_i^{\text{pix}})$
29: **end for**
30: $L_{\text{Cln}} \leftarrow \text{TripletLoss}\big(\{(e_i^{\text{cln}}, e_i^+)\}_{i=1}^B, \mu, \text{mine=batch-hardest-negative}\big)$
31: $L_{\text{AdvSem}} \leftarrow \text{TripletLoss}\big(\{(e_i^{\text{advSem}}, e_i^+)\}_{i=1}^B, \mu, \text{mine=batch-hardest-negative}\big)$
32: $L_{\text{AdvPix}} \leftarrow \text{TripletLoss}\big(\{(e_i^{\text{pix}}, e_i^+)\}_{i=1}^B, \mu, \text{mine=batch-hardest-negative}\big)$
33: $L \leftarrow \lambda_{\text{Cln}} L_{\text{Cln}} + \lambda_{\text{AdvSem}} L_{\text{AdvSem}} + \lambda_{\text{AdvPix}} L_{\text{AdvPix}}$          {$\lambda_{\text{Cln}} + \lambda_{\text{AdvSem}} + \lambda_{\text{AdvPix}} = 1$}
34: $\theta \leftarrow \theta - \eta \nabla_\theta L$
35: **return** $\theta$

---

## B  SELECTION OF ATTACKS' PERTURBATION BUDGETS

**Qualitative Examples** Figs. 5–6 provide qualitative examples of BOUNDSTYLE and DiffPrivate semantic attacks at different attack budgets. It can be seen that the original identities in the images become harder to identify as the attack budgets increase, leading to more aggressive semantic edits.

**User Study: Humans' Verification Accuracy** We conducted a user study with a convenience

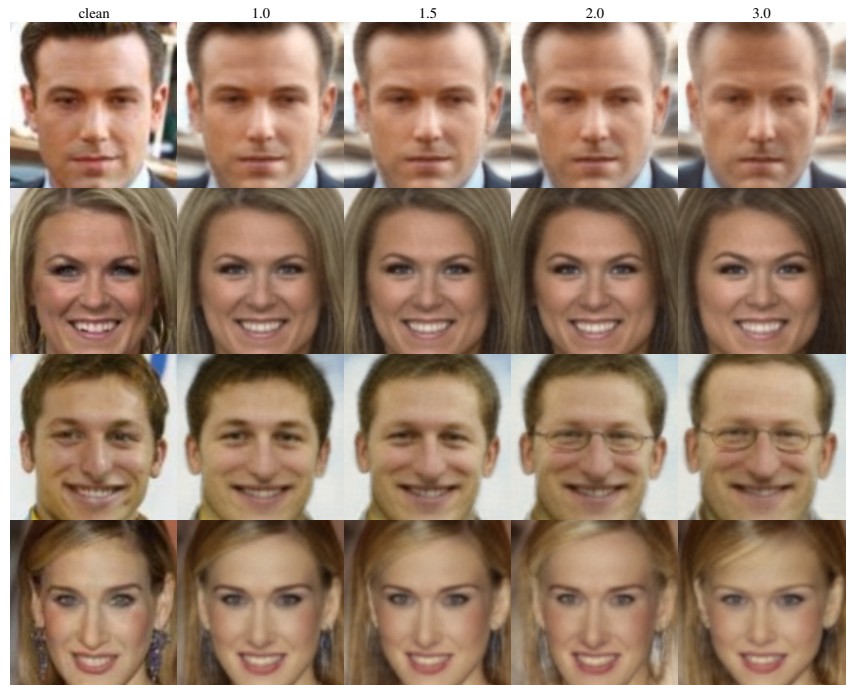

Figure 5: Examples of edits produced by BOUNDSTYLE when varying the attack budget $\beta \in \{1.0, 1.5, 2.0, 3.0\}$.

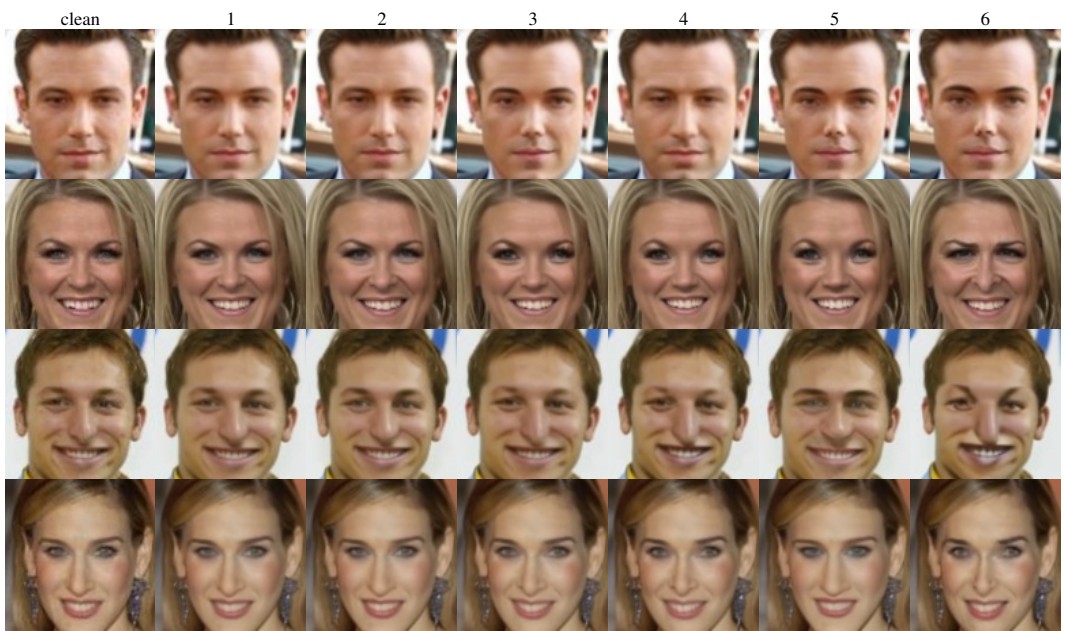

Figure 6: Examples of edits produced by DiffPrivate when varying the attack budget $\|\Delta z\| \in \{1, 2, 3, 4, 5, 6\}$.

sample ($N = 181$) to evaluate human recognition performance under different attack budgets. Our participants were asked to verify 30 randomly sampled image pairs each: 20 same-identity pairs where one image is clean and the other is either clean or adversarially modified by BOUNDSTYLE or DiffPrivate, and 10 different-identity pairs. The images were drawn from a combined pool of the LFW and VGG-Face datasets. For each presented pair, participants needed to judge whether the pairs depicted the same person (i.e., "same" or "different"). Fig. 7 details the results. We observe

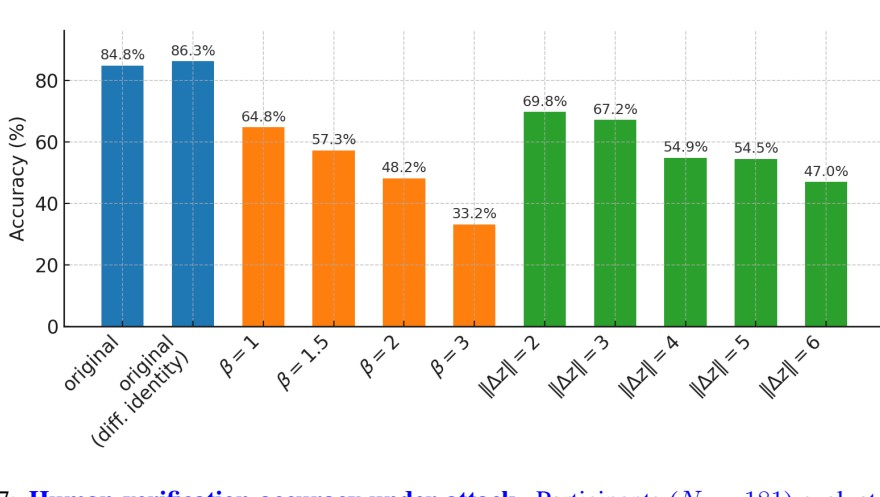

Figure 7: **Human verification accuracy under attack.** Participants ($N = 181$) evaluated image pairs at varying attack budgets. **Blue:** Original images (same and different identities); **Orange:** BOUNDSTYLE; **Green:** DiffPrivate.

that verification accuracy declines smoothly as the attack budgets ($\beta$ and $\|\Delta z\|$) increase, confirming these knobs control edit strength. At lower attack budgets ($\beta$=1 and $\|\Delta z\|$=2), participants maintain high accuracy (64.8% and 69.8%, respectively), retaining approximately 76–82% of the accuracy achieved on original clean pairs ($\approx 85\%$). Conversely, at the highest attack budgets we consider ($\beta$=3 and $\|\Delta z\|$=6), accuracy drops significantly (33.2% and 47%, respectively), roughly 39–55% of the clean images, justifying the capping of $\beta \leq 3$ and $\|\Delta z\| \leq 6$.

## C ABLATION STUDY

**Effect of $L_{\text{AdvPix}}$ During Training** Tab. 3 reports white-box robust accuracy on LFW for a ResNet model adversarially trained with and without $L_{\text{AdvPix}}$. Against DiffPrivate, enabling $L_{\text{AdvPix}}$ boosts robust accuracy substantially for high attack budgets, by up to 9.3%. Against BOUNDSTYLE, the effect of optimizing $L_{\text{AdvPix}}$ during training is minor and mixed, with -1.2–+0.93% difference in robust accuracy in comparison to when $L_{\text{AdvPix}}$ is not optimized. Overall, $L_{\text{AdvPix}}$ primarily helps ameliorate DiffPrivate at high attack budgets, while inducing only negligible changes against BOUNDSTYLE. Thus, we include this term by default in our training objective.

| | BoundStyle ($\beta$) | | | | | DiffPrivate ($\|\Delta z\|$) | | | | | |
| --- | --- | --- | --- | --- | --- | --- | --- | --- | --- | --- | --- |
| | 1 | 1.5 | 2 | 3 | | 1 | 2 | 3 | 4 | 5 | 6 |
| w/o $L_{\text{AdvPix}}$ | 98.61 | 90.74 | 83.80 | 51.87 | w/o $L_{\text{AdvPix}}$ | 99.54 | 99.07 | 97.69 | 79.63 | 46.30 | 37.96 |
| w/ $L_{\text{AdvPix}}$ | 97.69 | 91.67 | 84.26 | 50.72 | w/ $L_{\text{AdvPix}}$ | 99.54 | 99.54 | 97.69 | 86.57 | 54.17 | 47.22 |
| $\Delta$ | $-0.92$ | $+0.93$ | $+0.46$ | $-1.15$ | $\Delta$ | 0 | $+0.46$ | 0 | $+6.94$ | $+7.87$ | $+9.26$ |

Table 3: **The effect of $L_{\text{AdvPix}}$ on robust accuracy against white-box attacks.** We use LFW for evaluation.

**Attack Iterations** We analyze the impact of the number of attack iterations on the effectiveness of attacks. Tab. 4 shows the robust accuracy of the ResNet model on LFW under BOUNDSTYLE ($\beta = 3$) and DiffPrivate ($\|\Delta z\| = 5$) for varying numbers of iterations. For both attacks, we observe that attack success saturates after a certain number of steps. For BOUNDSTYLE, increasing iterations from 30 to 50 results in a marginal accuracy drop of only 0.76%. Similarly, for DiffPrivate, extending the attack from 70 to 250 iterations yields a negligible decrease of 0.47%. Based on these results, we fix the number of iterations to 30 for BOUNDSTYLE and 70 for DiffPrivate to balance attack strength with computational efficiency.

**Attacks' Time and Success Trade-offs** To further demonstrate that BOUNDSTYLE attains superior time and success-rate trade-offs compared to DiffPrivate, we execute both attacks under a matched

| BoundStyle ($\beta = 3$) | |
|---|---|
| Iterations | Accuracy (%) |
| 3 | 51.16 |
| 5 | 50.48 |
| 10 | 47.44 |
| 15 | 47.20 |
| **30** | **39.52** |
| 50 | 38.76 |

| DiffPrivate ($\|\Delta z\| = 5$) | |
|---|---|
| Iterations | Accuracy (%) |
| 10 | 98.15 |
| 50 | 51.85 |
| **70** | **41.67** |
| 100 | 41.20 |
| 250 | 41.20 |

Table 4: **Ablation on attack iterations.** We report robust accuracy on LFW against BOUNDSTYLE ($\beta = 3$) and DiffPrivate ($\|\Delta z\| = 5$) with varying iteration counts. The selected number of iterations is in boldface.

wall-clock constraint on the same hardware. Specifically, we run the attacks on an NVIDIA A6000 GPU with a limit of 6 seconds, which corresponds to approximately 30 iterations of BOUNDSTYLE. For DiffPrivate, we provide a significant advantage by removing caps on $\|\Delta z\|$ and the number of iterations, constraining it solely by the run time. Tab. 5 lists the results. It can be seen that, under equal execution time, DiffPrivate leaves robust accuracy at 98.6%, whereas BOUNDSTYLE degrades it to 85.2–50.9% (depending on $\beta$) against ResNet, on the LFW dataset. This result further highlights BOUNDSTYLE's efficiency and its ability to attain high success rates within strict time constraints, making it suitable for adversarial training.

| Attack | Budget ($\beta$ / $\|\Delta z\|$) | Time | Robust Acc. |
|---|---|---|---|
| DiffPrivate | $\infty$ | 6s | 98.6% |
| BOUNDSTYLE | 1.5 | 6s | 85.2% |
| BOUNDSTYLE | 2.0 | 6s | 75.0% |
| BOUNDSTYLE | 3.0 | 6s | 50.9% |

Table 5: **Comparing attacks' success, on LFW and ResNet, under matched time constraints (6 seconds).**

## D    IMPERCEPTIBLE PERTURBATIONS IN DIFFPRIVATE

We now present additional evidence that DiffPrivate introduces imperceptible perturbations. Recall that *(1)* DiffPrivate's success rates decrease significantly when applying semantics-preserving filters such as JPEG compression (Fig. 4 and Fig. 10); and that *(2)* adversarial training against PGD in the pixel space (employed in STYLEAT) improves robustness against DiffPrivate (Tab. 3). Both of these findings indicate that DiffPrivate introduce imperceptible perturbations besides semantic edits. To further investigate imperceptible perturbations produced by DiffPrivate, we analyze the frequency-energy patterns of the adversarial perturbations (i.e., the pixel-wise difference between original and adversarial images). To this end, we convert the perturbations produced by attacks (i.e., difference between edited and clean images) to the Fourier domain and measure the cumulative energy outside an increasing radial distance $r$ from the zero frequency (i.e., DC) component. The result is monotonic curves decreasing from 1 to 0, as shown in Fig. 8, which illustrate the energy distribution: a faster decay indicates energy concentrated in low frequencies (i.e., semantic changes), while a slower decay implies reliance on high frequencies. We find that the decay is significantly faster for BOUNDSTYLE ($> 90\%$ of residual energy contained within $r \approx 15$) than for DiffPrivate ($>90\%$ of residual energy within $r \approx 30$). This results further demonstrates that DiffPrivate's success can be partially attributed to non-semantic, imperceptible perturbations.

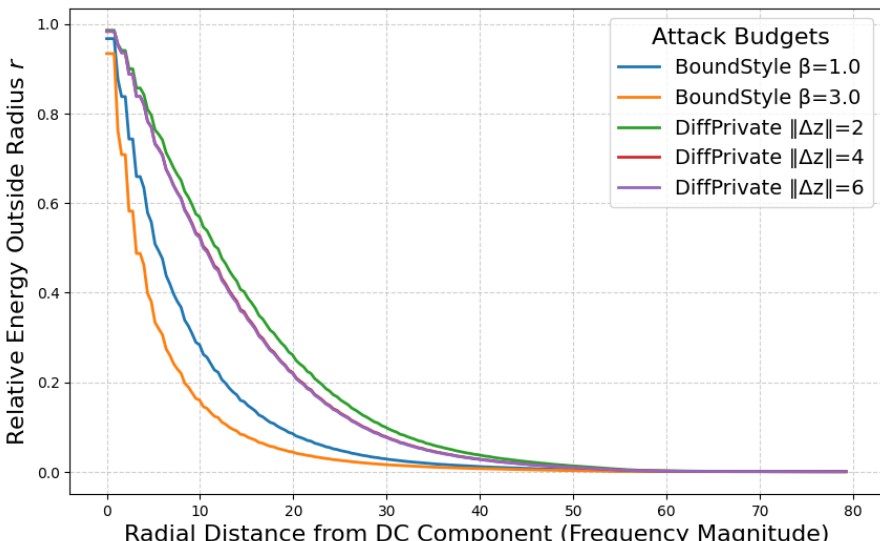

Figure 8: **Frequency energy decay of adversarial perturbations.** We measure the relative energy of the noise residual outside frequency radius $r$. BOUNDSTYLE (orange/blue) decays rapidly, indicating changes are concentrated in low frequencies (semantic). DiffPrivate (green/red/purple) decays slower, indicating a reliance on higher frequencies (imperceptible noise). Results are shown for the red channel of LFW, but we observe consistent behavior across all color channels.

# E GRAY-BOX ATTACKS AGAINST REPVGG TRAINED WITH STYLEAT

We present the gray-box evaluation results for the RepVGG backbone in Figs. 9–10. Consistent with the ResNet findings, STYLEAT demonstrates superior robustness compared to filter-based defenses. For instance, against DiffPrivate attack on the LFW dataset (at $\|\Delta z\| = 6$), STYLEAT achieves a robust accuracy of 88.9%, significantly outperforming the strongest filter defense (Feature Squeezing at 69.9%) and the undefended model (38.0%). Similar trends are observed against the BOUND-STYLE attack, where filter-based defenses fail to provide meaningful robustness gains compared to STYLEAT.

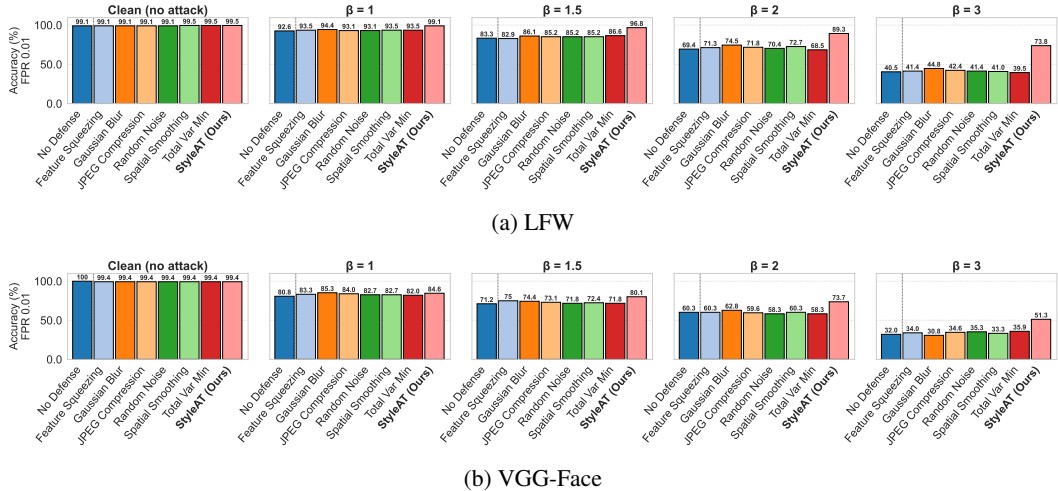

Figure 9: **Evaluating defenses against gray-box BOUNDSTYLE attacks (RepVGG).**

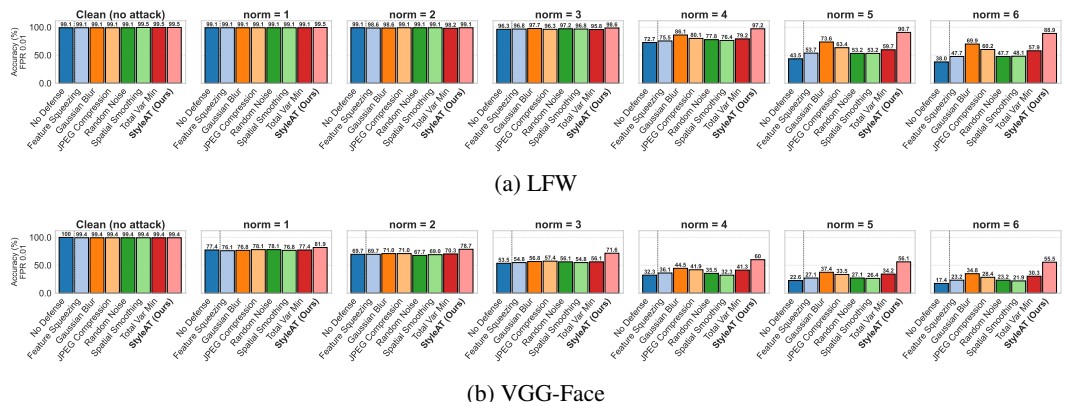

(a) LFW

(b) VGG-Face

Figure 10: **Evaluating defenses against gray-box DiffPrivate attacks (RepVGG).**

# F    SEMANTIC STRUCTURE OF BOUNDSTYLE ADVERSARIAL DIRECTIONS

To explore how BOUNDSTYLE manipulates distinct semantic attributes, we analyze the nature of the adversarial perturbations. To do so, we collect the latent perturbations $\delta$ generated by BOUNDSTYLE (at $\beta = 3$) against the ResNet backbone on the LFW dataset. Subsequently, we then perform Principal Component Analysis (PCA) on these perturbation vectors to identify the dominant directions of variance in the attack space.

Fig. 11 visualizes the top five principal components (PCs). To interpret these abstract vectors, we project the perturbation $\delta_i$ of each test sample $i$ onto each PC. Then, for every PC, we select the top five samples with the highest positive projection scores, i.e., the images whose adversarial manipulations align most strongly with that specific principal direction. We visualize examples with $\beta=3$ to maximally emphasize the semantic nature of the directions. Our visual analysis demonstrates that the dominant modes of the attack correspond to coherent semantic factors: **PC1** captures aging (older appearance, thinning hair, and beard growth); **PC2** modifies lighting and alters nose shape; **PC3** creates a younger appearance by smoothing skin texture and softening facial features; **PC4** alters head pose; and **PC5** performs subtle structural changes to facial width. This analysis confirms that BOUNDSTYLE discovers and exploits interpretable semantic weaknesses in the target FR model.

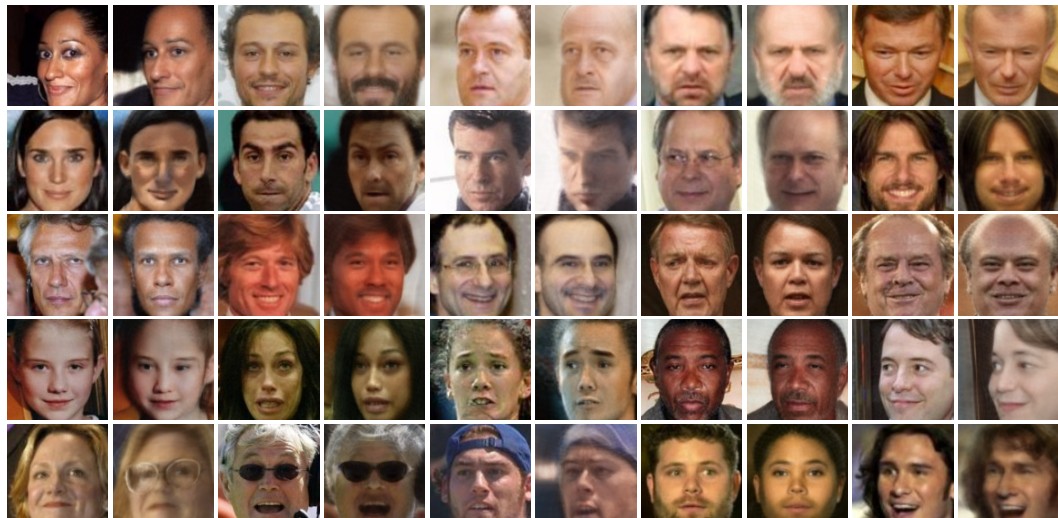

Figure 11: Visualization of image samples whose adversarial perturbations align most strongly with the top five Principal Components (PCs) of the BOUNDSTYLE perturbation distribution. Each column pair displays an original (left) and an adversarial (right) image. The dominant directions correspond to clear semantic attributes: **PC1** induces aging effects (older appearance); **PC2** alters nose shape and illumination; **PC3** creates a younger appearance; **PC4** corresponds to head pose adjustments; and **PC5** performs subtle structural changes to facial width.

## USE OF LARGE LANGUAGE MODELS (LLMS)

We used ChatGPT to assist in polishing some text and help improve the quality of the presentation. All outputs of the LLM have been manually verified and most have been further edited by the authors. The analyses, ideas, and technical content, of course, are all original.

