# OpenReview forum: "StyleAT: Defending Face Recognition Against Semantic Attacks"
_ICLR.cc/2026/Conference — Submitted to ICLR 2026_

### Official Review · Reviewer_xjDT · 2025-10-27

**Soundness:** 2
**Presentation:** 3
**Contribution:** 2
**Rating:** 2
**Confidence:** 4

**Summary:**

The paper introduces BOUNDSTYLE, a semantic adversarial attack that manipulates StyleGAN3 latent codes to generate perturbed face images capable of evading face recognition (FR) systems while preserving perceptual identity for human observers. Additionally, the authors propose STYLEAT, an adversarial training framework that incorporates three distinct triplet losses computed on clean images, BOUNDSTYLE-edited samples, and PGD pixel-space perturbations to enhance model robustness against semantic attacks. Experimental evaluation across five FR architectures and two benchmark datasets demonstrates that BOUNDSTYLE exhibits strong transferability across different recognizers while achieving approximately 9.5× faster generation speed compared to DiffPrivate.

**Strengths:**

The paper presents BOUNDSTYLE, an efficient StyleGAN3 latent-space attack that is approximately 9.5× faster than DiffPrivate, enabling practical adversarial training at scale.

Building on this efficiency, the authors propose STYLEAT, which combines clean, semantic-attack, and PGD pixel-level triplet losses to improve robustness across five face recognition models and two unseen datasets.

**Weaknesses:**

1. The paper uses “semantics” as edits to expression/age/accessories with identity preserved, but provides no quantitative disentanglement or taxonomy of which semantic factors are actually being manipulated and protected.
2. At line 62, the authors claim that BOUNDSTYLE produces only semantic edits; however, the paper lacks both qualitative visualizations and quantitative validation to substantiate this claim.
3. This work represents an incremental contribution with limited novelty. The methodology is quite unclear, and without proper architecture, it is very difficult to follow. A complete end-to-end pipeline might help.
4. The attack objective (Section 4.1) uses the recognizer on preprocessed crops of generated images, i.e., F(C(G(l+δ)). Applying crop/alignment after editing may itself alter pixel evidence and confound what counts as a “semantic” change.
5. All target FR backbones are classic CNNs with one Swin-T. No recent SOTA recognizers (e.g., MagFace, AdaFace) are included, weakening robustness and generalizability claims.
6. The comparison presented is methodologically flawed, as it evaluates results using β = 3 (BOUNDSTYLE or STYLEAT) against z=6 for DiffPrivate. This creates an unfair benchmark since DiffPrivate demonstrates superior robustness at z=3, which would be a more appropriate comparison point for evaluating the proposed approaches.
7. The paper does not showcase statistical testing.
8. The paper claims high visual fidelity through inversion and pivotal tuning, asserting that edits appear unchanged to human observers. However, no quantitative perceptual metrics (SSIM, PSNR, LPIPS) were provided, nor were any human evaluation studies conducted to support these claims.

**Questions:**

(1) How does the paper quantitatively measure and verify that BOUNDSTYLE manipulates only semantic attributes (expression, age, accessories) while preserving identity? Can authors provide a systematic taxonomy or disentanglement analysis of which semantic factors are being modified and protected?

(2) Regarding the claim at line 62 that BOUNDSTYLE produces only semantic edits, can authors provide any qualitative visualizations or quantitative metrics to validate this assertion?

(3) Could authors provide a comprehensive end-to-end architectural diagram or pipeline illustration to clarify the methodology? The current presentation makes it difficult to understand the complete workflow and implementation details.

(4) In Section 4.1, the attack objective applies face recognition to preprocessed crops of generated images (F(C(G(l+δ)))). How does this account for the fact that post-generation cropping and alignment may introduce pixel-level artifacts that confound the distinction between semantic and identity-related changes?

(5) The proposed method should be validated against recent state-of-the-art face recognizers such as MagFace and AdaFace. How would their inclusion affect the robustness and generalizability claims, given that the current evaluation relies primarily on classic CNN architectures with only one Transformer model (Swin-T)?

(6)  Can the author justify the choice of comparing β=3 for your methods against z=6 for DiffPrivate? Since DiffPrivate demonstrates superior robustness at z=3, wouldn't this be a fairer and appropriate baseline for comparison?

(7) The paper should include statistical significance tests to validate that the reported performance improvements are statistically meaningful rather than within noise margins.

(8) The claim of high visual fidelity asserts that edits appear unchanged to human observers. Can the author provide quantitative perceptual quality metrics (SSIM, PSNR, LPIPS) and conduct human evaluation studies to substantiate these claims?

---

> ### Author Response · Authors · 2025-11-22
>
> We thank the reviewer for the constructive comments and address their concerns in what follows (a revised PDF will follow soon).
> ### Budget mapping (β vs. ‖Δz‖)
> The attack budgets live in different generative manifolds and modulate different phenomena. Primarily, we find that DiffPrivate is not purely semantic (see the next point): beyond editing attribute, it introduces non-semantic, high-frequency residuals. Hence, attack budgets that equally alter semantics may not correspond to equivalent attack strengths.
> ### Imperceptible perturbations in DiffPrivate
> Our submission presents strong evidence that DiffPrivate introduces imperceptible perturbations: (1) DiffPrivate’s success rates significantly decrease when applying semantics-preserving filters (e.g., JPEG compression); and (2) Adversarial training against PGD in the pixel space (as part of StyleAT) improves robustness against the attack (Table 3). As additional support, we analyzed frequency-energy patterns in adversarial perturbations (i.e., pixel-wise differences between the original and adversarial images), per reviewers’ suggestions. Specifically, we converted perturbations to the Fourier domain and measured the cumulative energy outside an increasing radius $r$; the result is monotonic curves decreasing from 1 to 0, where a faster decay indicates that energy is heavily concentrated in low frequencies (i.e., semantic changes). Indeed, we find that the decay is faster for BoundStyle (>90% of residual energy within $r\approx{}15$) than for DiffPrivate (>90% of residual energy within $r\approx{}30$), further demonstrating that DiffPrivate’s success can be partially attributed to non-semantic, imperceptible perturbations.
> ### Time and success-rate trade-offs
> To further demonstrate that BoundStyle attains superior time and success-rate trade-offs that DiffPrivate, we ran the attacks under a matched wall-clock on the same device. Specifically, we executed attacks on an Nvidia A6000 GPU for 6 seconds, roughly the time needed to complete 30 iterations of BoundStyle. We also gave DiffPrivate advantage, by not capping  ∥Δz∥ or the number of iterations. In this setting, under equal run time, DiffPrivate leaves robust accuracy at 98.6%, whereas BoundStyle degrades it to 85%—51% (depending on $\beta$) against ResNet, on the LFW dataset.
> ### User study
> For qualitative evaluation of both BoundStyle and DiffPrivate, please see Figure 4 in the paper. For quantitative validation, we conducted a user study with a convenience sample (N=153) to test whether humans can recognize subjects after applying attacks. We randomly sampled 30 image pairs: 20 same-identity pairs where one image is clean and the other is either clean or edited with BoundStyle or DiffPrivate, and 10 different-identity pairs. The participants were asked to judge pairs a same or different. We found that participants’ accuracy declined smoothly with β and ‖Δz‖, confirming these knobs control edit strength. At low attack budgets (β=1 and ‖Δz‖=1), participants had high accuracy, nearing ~75% of the accuracy on clean images. In contrast, at high budgets (β=3 and ‖Δz‖=6), participants’ accuracy was about $\sim$50% of their accuracy on clean images, justifying why we capped these two parameters to 3 and 6, respectively.
> ### Additional state-of-the-art recognizers
> Beyond the four CNNs and Swin Transformer already covered in the paper, we evaluated attacks against MagFace and ArcFace (MobileFaceNet backbones) in white- and black-box settings. Across both datasets, they behave like the other FR models: robust accuracy drops smoothly with β, and off-diagonals are close to diagonals (the attack transfers). E.g., on LFW, the BoundStyle’s white-box robust accuracies at β=3 are 45% (ArcFace) and 52% (MagFace), while black-box robust accuracies across sources fall in the 57%–72% range. These trends mirror our other backbones and support our robustness/transferability conclusions.
> ### Disentanglement analysis
> We appreciate this suggestion and plan to conduct the proposed analysis.
> ### Cropping
> We intentionally evaluate the recognizer with the same preprocessing used during training. Omitting (or changing) the cropping induces a distribution shift that hurts clean accuracy regardless of any semantic edit, thus confounding the measurements.
> ### Diagrams
> We concur that overview figures can improve the clarity of our presentation. We have created two diagrams depicting how BoundStyle and StyleAT work and will include them in the updated PDF.

---

> > ### Comment · Reviewer_xjDT · 2025-11-24
> > **Discussion**
> >
> > Thanks for the responses. Can the authors point out where the responses to the questions raised are provided in the rebuttal text or the PDF?
> >
> > The response to the majority of the concerns (Q. 1-3) is not clearly visible. While the concerns about other face recognisers are partially addressed in the text, they are neither detailed nor presented in the paper. The same is with the Q. 8 asked in the
> > the "Questions" section of the initial reviews.
> >
> > Thanks

---

### Official Review · Reviewer_squ7 · 2025-10-30

**Soundness:** 2
**Presentation:** 1
**Contribution:** 2
**Rating:** 4
**Confidence:** 3

**Summary:**

This paper studies semantic adversarial attacks on face recognition. It introduces a new attack called **BOUNDSTYLE**, which operates in the latent space of StyleGAN3 to produce realistic facial edits that fool recognition systems.  BOUNDSTYLE is efficient, tunable, and achieves high attack success with realistic outputs.  Building on it, the authors propose **STYLEAT**, an adversarial training method that uses fast BOUNDSTYLE variants to improve robustness.  STYLEAT combines latent-space semantic attacks and small pixel-space perturbations during training.  Experiments on **LFW** and **VGG-Face** show that BOUNDSTYLE is about 9.5× faster than the diffusion-based DiffPrivate.  The paper claims that STYLEAT is the first defense specifically designed for general semantic attacks on face recognition.  The experiments are carried out on two datasets (LFW and VGG-Face) using pre-trained face recognition backbones.

**Strengths:**

-   **Clear methodological contribution:** The paper introduces both a new semantic attack (BOUNDSTYLE) and a corresponding defense (STYLEAT), forming a coherent attack–defense framework.

-   **Practical efficiency:** BOUNDSTYLE is nearly an order of magnitude faster than prior semantic attacks, making it feasible for adversarial training and large-scale robustness evaluation

**Weaknesses:**

-   **No high-level block diagram of the pipeline.**
    The paper provides a clear pseudocode (Alg. 1 in the appendix) but lacks a single illustration or block diagram showing the end-to-end attack → defense training pipeline, which makes the method harder to digest quickly.

-   **No analytic mapping between attack budgets in different latent spaces.**
    BOUNDSTYLE’s budget (β in StyleGAN3 latent space) and DiffPrivate’s budget (‖Δz‖ in diffusion latent space) are treated as separate knobs without a principled conversion. The paper therefore compares attacks empirically but provides no theory or metric to equate perceptual magnitudes across the two spaces.

-   **Transferability claim is overstated relative to absolute potency.**
    BOUNDSTYLE shows _good transferability_ (off-diagonals close to diagonals), but DiffPrivate drives _lower absolute_ robust accuracies in some settings (notably VGG-Face), meaning DiffPrivate can be a more destructive white-box attack even if it transfers less well. The paper reports both effects but leans on transferability as a strength without emphasizing the potency tradeoff enough.

-   **Baselines, datasets, and backbone coverage are limited.**
    DOA and simple filters are not fully representative baselines for latent-space semantic attacks (DOA targets ad-hoc physical accessories), and STYLEAT is adversarially trained with BOUNDSTYLE, which naturally advantages it against that family of edits. The experimental scope is also narrow: evaluation is on two datasets (LFW, VGG-Face) using pre-trained backbones, without experiments on larger or cleaner FR datasets (e.g., VGGFace2, Glint360K) or broader transformer backbones. These choices weaken claims of broad generalization.

**Questions:**

-   **Computation claim:** Could the authors provide a FLOP or compute-based analysis to support the reported 9.5× speedup?

-   **Other concerns:** Please refer to the weaknesses section for additional points regarding clarity, evaluation setup, and baseline choices.

---

> ### Author Response · Authors · 2025-11-22
>
> We thank the reviewer for the constructive comments and address their concerns in what follows (a revised PDF will follow soon).
> ### Time and success-rate trade-offs
> To further demonstrate that BoundStyle attains superior time and success-rate trade-offs that DiffPrivate, we ran the attacks under a matched wall-clock on the same device. Specifically, we executed attacks on an Nvidia A6000 GPU for 6 seconds, roughly the time needed to complete 30 iterations of BoundStyle. We also gave DiffPrivate advantage, by not capping  ∥Δz∥ or the number of iterations. In this setting, under equal run time, DiffPrivate leaves robust accuracy at 98.6%, whereas BoundStyle degrades it to 85%—51% (depending on $\beta$) against ResNet, on the LFW dataset.
> ### Budget mapping (β vs. ‖Δz‖)
> The attack budgets live in different generative manifolds and modulate different phenomena. Primarily, we find that DiffPrivate is not purely semantic: beyond editing attribute, it introduces non-semantic, high-frequency residuals. Hence, attack budgets that equally alter semantics may not correspond to equivalent attack strengths. We support our hypothesis two observations included in the paper: (1) DiffPrivate’s success rates significantly decrease when applying semantics-preserving filters (e.g., JPEG compression); and (2) Adversarial training against PGD in the pixel space (as part of StyleAT) improves robustness against the attack (Table 3). As additional support, we analyzed frequency-energy patterns in adversarial perturbations (i.e., pixel-wise differences between the original and adversarial images), per reviewers’ suggestions. Specifically, we converted perturbations to the Fourier domain and measured the cumulative energy outside an increasing radius $r$; the result is monotonic curves decreasing from 1 to 0, where a faster decay indicates that energy is heavily concentrated in low frequencies (i.e., semantic changes). Indeed, we find that the decay is faster for BoundStyle (>90% of residual energy within $r\approx{}15$) than for DiffPrivate (>90% of residual energy within $r\approx{}30$), further demonstrating that DiffPrivate’s success can be partially attributed to non-semantic, imperceptible perturbations.
> ### Defense baselines
> To our knowledge, StyleAT is the first defense explicitly designed to deter general semantic attacks on FR (i.e., we are not aware of established defenses against this class of attacks to compare against). Accordingly, we structure evaluation around two orthogonal attack families: BoundStyle (GAN-based) and DiffPrivate (diffusion-based). As a defense baseline, DOA is informative: it performs well against DiffPrivate yet does not defend against BoundStyle (and may slightly degrade robustness), showing StyleAT is not simply exploiting an easy baseline.
> ### Dataset choice
> We intentionally use the LFW and VGG-Face datasets in our evaluations for multiple reasons:
> * Near-perfect clean accuracy on our FR backbones allows us to attribute any drop directly to semantic edits, and recovery to the defense.
> * Consistency with prior work (e.g., DiffPrivate).
> * Running the evaluations on these datasets with multiple models and attack configurations is tractable given our resource constraints.
> ### Additional state-of-the-art recognizers
> Beyond the four CNNs and Swin Transformer already covered in the paper, we evaluated attacks against MagFace and ArcFace (MobileFaceNet backbones) in white- and black-box settings. Across both datasets, they behave like the other FR models: robust accuracy drops smoothly with β, and off-diagonals are close to diagonals (the attack transfers). E.g., on LFW, the BoundStyle’s white-box robust accuracies at β=3 are 45% (ArcFace) and 52% (MagFace), while black-box robust accuracies across sources fall in the 57%–72% range. These trends mirror our other backbones and support our robustness/transferability conclusions.
> ### Diagrams
> We concur that overview figures can improve the clarity of our presentation. We have created two diagrams depicting how BoundStyle and StyleAT work and will include them in the updated PDF.

---

### Official Review · Reviewer_Bek9 · 2025-10-31

**Soundness:** 3
**Presentation:** 4
**Contribution:** 3
**Rating:** 6
**Confidence:** 4

**Summary:**

This paper addresses the issue of face recognition (FR) systems being vulnerable to semantic attacks (e.g., changes in age or pose). Existing attacks are either computationally costly or have limited effectiveness, making it difficult to develop effective defenses. Furthermore, there is a lack of established defenses against general semantic attacks.
To address these issues, this paper presents two main contributions:
BOUNDSTYLE: A novel and efficient semantic attack method that operates in StyleGAN's latent space. Experiments show that while maintaining a high attack success rate, it is approximately 9.5 times faster than state-of-the-art (SOTA) attacks like DiffPrivate , making it well-suited for adversarial training.
STYLEAT: An adversarial training defense scheme based on BOUNDSTYLE. This scheme enhances model robustness using a low-cost variant of BOUNDSTYLE , and it also integrates a component to defend against pixel-level perturbations.
The experimental evaluation was conducted on two datasets (LFW and VGG-Face) and five FR models. The results show that STYLEAT significantly improves the model's robust accuracy against SOTA attacks (including DiffPrivate, which was unseen during training), with up to a 46.3% increase in the gray-box setting, for example , and it outperforms existing defense methods (such as DOA and standard filters)。

**Strengths:**

This paper's strengths lie in its dual contribution: it introduces both an efficient, powerful semantic attack (BOUNDSTYLE) and an effective, generalizable defense (STYLEAT). A significant advantage is its solution to the feasibility problem of adversarial training; BOUNDSTYLE's high efficiency (about 9.5 times faster than DiffPrivate ) directly overcomes the high computational cost that previously made such defenses impractical. The STYLEAT defense demonstrates strong efficacy and generalization, proving effective not only against the BOUNDSTYLE attack used in its training but also generalizing to unseen SOTA attacks like DiffPrivate, with robust accuracy improving up to 28.6% 6and 46.3%, respectively. Moreover, BOUNDSTYLE itself is a potent attack, matching DiffPrivate in white-box scenarios while exhibiting clearly superior black-box transferability. Finally, the paper provides a key design insight by incorporating LAdvPix, based on the observation that some attacks like DiffPrivate may include imperceptible pixel perturbations , a hypothesis confirmed by ablation studies.

**Weaknesses:**

1.Limitation of Defense Evaluation: Although the attack (BOUNDSTYLE) was tested on 5 different FR backbones 1, the defense's (STYLEAT) effectiveness appears to be evaluated only on the ResNet model.
2.Empirical Defense: As the authors acknowledge in the conclusion, STYLEAT is an empirical defense and does not provide provable security guarantees8。

**Questions:**

1. Why was the STYLEAT defense only evaluated on the ResNet backbone ? Given that the attack tests covered multiple architectures (such as SwinT), showing STYLEAT's defensive effectiveness on these different architectures (particularly its cross-architecture generalization) would make the paper's conclusions more convincing？

---

> ### Author Response · Authors · 2025-11-22
>
> We thank the reviewer for the constructive comments and address their concerns in what follows (a revised PDF will follow soon).
> ### SyleAT on additional models
> To validate the generality of our findings, we also trained StyleAT on FaceX-Zoo’s VGG model. Across all evaluation datasets (LFW and VGG-Face), attacks (BoundStyle and DiffPrivate), and settings (i.e., gray- and white-box) considered, we observed similar trends to those found on ResNet (e.g., +30% and +14 robust accuracy in the gray- and white-box settings, respectively, against BoundStyle with $\beta$=3 on LFW).
> ### Additional state-of-the-art recognizers
> We evaluated attacks against MagFace and ArcFace (MobileFaceNet backbones) in white- and black-box settings. Across both datasets, they behave like the other FR models: robust accuracy drops smoothly with β, and off-diagonals are close to diagonals (the attack transfers). E.g., on LFW, the BoundStyle’s white-box robust accuracies at β=3 are 45% (ArcFace) and 52% (MagFace), while black-box robust accuracies across sources fall in the 57%–72% range. These trends mirror our other backbones and support our robustness/transferability conclusions.
> ### Empirical defense
> We agree that StyleAT constitutes an empirical approach for improving robustness against general semantic attacks. Our aim is to provide a practical method that is effective against both attacks known during training time (BoundStyle) as well as held-out attacks (DiffPrivate), consistent across backbones (ResNet and RepVGG), and efficient. Prior work on adversarial training has shown this paradigm to be durable, withstanding the test of time (chiefly, see Madry et al.).

---

### Official Review · Reviewer_dRaU · 2025-11-02

**Soundness:** 2
**Presentation:** 3
**Contribution:** 2
**Rating:** 4
**Confidence:** 4

**Summary:**

The paper introduces a  StyleGAN3-latent, tunable semantic attack on face recognition (BOUNDSTYLE) and an adversarial-training scheme that mixes fast BOUNDSTYLE variants with light PGD in pixel space to improve robustness against general semantic attacks (STYLEAT).  Experiments span five FR backbone networks and are evaluated on two well-known face recognition datasets. The results demonstrate BOUNDSTYLE exceeds/matches a norm-bounded baseline by about 10x faster. STYLEAT achieves better white-box robustness.

**Strengths:**

1. This paper focuses on semantic attacks, which are more practical than norm-bounded noise. It launches tunable and fast attacks and achieves about 10x faster.

2. Proposed defense improves robustness to unseen attack in white- and gray-box settings. Black-box heatmaps show proposed attacking methods transfer across architectures more than the baseline.

**Weaknesses:**

1. Robustness is reported only on LFW and VGG-Face with limited positive pairs at a fixed FPR target; these are aging, small-scale verification sets. No tests on harder, modern FR benchmarks (e.g., IJB-C/IJB-S, MegaFace reruns) or large-scale ID/verification datasets.
2. The proposed attack - STYLEAT is applied to ResNet only; robustness gains may not generalize to other strong backbones (e.g., ArcFace-like ResNet100, ViT-based SOTA).
3. The paper qualitatively limits β/‖Δz‖ to “preserve identity,” but provides no human study or perceptual metric (e.g., face-ID match rate against a separate high-accuracy verifier, or MOS/LPIPS) to confirm “same person” under edits.
4. Baseline DiffPrivate is evaluated as a norm-bounded variant with capped iterations (∥Δz∥≤6, 70 iters), which may understate its best-known strength; there’s no “best-effort” setting matching wall-clock or quality across methods.
5. Results emphasize robust accuracy at a fixed FPR on small test sets; no ROC tradeoffs, no calibration robustness, and no analysis under different thresholds or across demographics/poses.  The 9.5× speedup is measured with batch=1 on a single A6000, without reporting attack convergence/tuning across GPUs or batched/mixed-precision settings; unclear generality. STYLEAT includes pixel-space PGD partly to counter “imperceptible perturbations” in DiffPrivate, inferred via filter sensitivity. This is indirect evidence; more direct spectral/energy analyses would strengthen the claim.

**Questions:**

1. (Important) How do you quantify that BOUNDSTYLE edits preserve identity? Please report verification rates against a strong external FR (not used in training or thresholding), plus perceptual metrics (LPIPS/FID) or a small human study stratified by β.

2. Could the authors add a matched wall-clock comparison (equal time budgets) and a best-effort comparison (no norm cap; tuned steps) for DiffPrivate vs. BOUNDSTYLE, to remove confounds from ∥Δz∥ bounds and iteration caps? Do the robustness gains transfer when defending SwinT/MobileFace/RepVGG (and newer ArcFace-style models)? Any signs of robust overfitting across backbones or datasets?

3. How do results change if you (a) vary the FPR target (e.g., 1e-3, 1e-5), (b) recalibrate post-training, and (c) plot full ROC/DET curves under attack?   Can the authors report results on harder, modern FR benchmarks (IJB-C/S, CFP-FP, AgeDB, CALFW, CPLFW) and larger positive/negative sets to reduce variance?

4. Beyond filter sensitivity, do the authors have frequency-domain or gradient-alignment analyses showing DiffPrivate introduces non-semantic high-frequency components, and that STYLEAT’s pixel PGD directly targets them? Or if this is inapplicable to the current method?

5. Please ablate T, β, α for BOUNDSTYLE during training vs testing, report compute/accuracy trade-offs, and show whether per-sample random starts are essential. Inversion and pivotal tuning are often brittle. Which encoder, learning rates, and stopping criteria were used? What cache reuse ratio and wall-clock overhead do they add to training?

---

> ### Author Response · Authors · 2025-11-22
>
> We thank the reviewer for the constructive comments and address their concerns in what follows (a revised PDF will follow soon).
> ### SyleAT on additional models
> To validate the generality of our findings, we also trained StyleAT on FaceX-Zoo’s VGG model. Across all evaluation datasets (LFW and VGG-Face), attacks (BoundStyle and DiffPrivate), and settings (i.e., gray- and white-box) considered, we observed similar trends to those found on ResNet (e.g., +30% and +14 robust accuracy in the gray- and white-box settings, respectively, against BoundStyle with $\beta$=3 on LFW).
> ### Imperceptible perturbations in DiffPrivate
> Our submission presents strong evidence that DiffPrivate introduces imperceptible perturbations: (1) DiffPrivate’s success rates significantly decrease when applying semantics-preserving filters (e.g., JPEG compression); and (2) Adversarial training against PGD in the pixel space (as part of StyleAT) improves robustness against the attack (Table 3). As additional support, we analyzed frequency-energy patterns in adversarial perturbations (i.e., pixel-wise differences between the original and adversarial images), per reviewers’ suggestions. Specifically, we converted perturbations to the Fourier domain and measured the cumulative energy outside an increasing radius $r$; the result is monotonic curves decreasing from 1 to 0, where a faster decay indicates that energy is heavily concentrated in low frequencies (i.e., semantic changes). Indeed, we find that the decay is faster for BoundStyle (>90% of residual energy within $r\approx{}15$) than for DiffPrivate (>90% of residual energy within $r\approx{}30$), further demonstrating that DiffPrivate’s success can be partially attributed to non-semantic, imperceptible perturbations.
> ### Time and success-rate trade-offs
> To further demonstrate that BoundStyle attains superior time and success-rate trade-offs that DiffPrivate, we ran the attacks under a matched wall-clock on the same device. Specifically, we executed attacks on an Nvidia A6000 GPU for 6 seconds, roughly the time needed to complete 30 iterations of BoundStyle. We also gave DiffPrivate advantage, by not capping  ∥Δz∥ or the number of iterations. In this setting, under equal run time, DiffPrivate leaves robust accuracy at 98.6%, whereas BoundStyle degrades it to 85%—51% (depending on $\beta$) against ResNet, on the LFW dataset.
> ### Dataset choice
> We intentionally use the LFW and VGG-Face datasets in our evaluations for multiple reasons:
> * Near-perfect clean accuracy on our FR backbones allows us to attribute any drop directly to semantic edits, and recovery to the defense.
> * Consistency with prior work (e.g., DiffPrivate).
> * Running the evaluations on these datasets with multiple models and attack configurations is tractable given our resource constraints.
> ### User study
> We conducted a user study with a convenience sample (N=153) to test whether humans can recognize subjects after applying attacks. We randomly sampled 30 image pairs: 20 same-identity pairs where one image is clean and the other is either clean or edited with BoundStyle or DiffPrivate, and 10 different-identity pairs. The participants were asked to judge pairs a same or different. We found that participants’ accuracy declined smoothly with β and ‖Δz‖, confirming these knobs control edit strength. At low attack budgets (β=1 and ‖Δz‖=1), participants had high accuracy, nearing ~75% of the accuracy on clean images. In contrast, at high budgets (β=3 and ‖Δz‖=6), participants’ accuracy was about $\sim$50% of their accuracy on clean images, justifying why we capped these two parameters to 3 and 6, respectively.
> ### Parameter selection
> * _Run-time comparisons_: we set the batch size to one because the public DiffPrivate implementation does not support larger batch sizes.
> * _Attack iterations_: We set the number of iterations to 30 and 70 for BoundStyle and DiffPrivate, respectively, as increasing these leads to little-to-no improvement in the attacks’ success rates.
> * _Step size_: We use α=β in BoundStyle throughout our evaluations for consistency with StyleAT; preliminary experiments show that tuning α only makes the improvement gains on StyleAT more significant, thus strengthening our findings.

---

> > ### Comment · Reviewer_dRaU · 2025-11-25
> >
> > Thank you for the detailed rebuttal and the additional experiments.
> >
> > The frequency-domain analysis comparing BoundStyle and DiffPrivate is helpful. The matched wall-clock experiment (6-second budget on A6000, with uncapped DiffPrivate iterations) partially addresses my request for equal-time and best-effort comparisons.  The user study (N=153) provides perceptual evidence regarding identity preservation and the effect of β/‖Δz‖ on recognizability. This addresses my core concern about the lack of perceptual validation or external FR verification for identity preservation.
> >
> > While the rationale for using LFW and VGG-Face is noted, these datasets remain small and aging. My concern regarding the lack of experiments on modern, large-scale benchmarks. Adding VGG is a positive step, but the core defense (StyleAT) has not yet been evaluated on strong, modern FR backbones such as ArcFace-style ResNet100, MobileFace, Swin/ViT-based FR, or RepVGG. It remains unclear whether robustness gains generalize or whether StyleAT suffers from robust overfitting to specific architectures. The rebuttal does not address my requests for evaluations at very low false-positive rates (1e-3, 1e-5) or for full ROC/DET curves. This is important because robustness at a single operating point on small verification sets can be misleading, and practical FR deployments typically operate at strict FPRs. Some hyperparameter rationales were provided, but full ablations are still missing.
> >
> > Overall, the rebuttal addresses several of the technical questions and strengthens the evidence on frequency characteristics, etc.  However, the major concerns regarding benchmark breadth, backbone diversity, robustness evaluation at strict FPRs, and calibration/threshold analysis remain unresolved. For these reasons, my original score remains unchanged.

---

### Author Response · Authors · 2025-12-03
**General final response to reviewers**

We thank all reviewers for their constructive comments and the time spent evaluating our work. In response to the collective feedback, we revised our paper to strengthen our claims and address the comments. To facilitate a quick review of these updates, we have highlighted all major changes in blue within the revised PDF. Below is a summary of the primary contributions added in this revision:

### Expanded Model Evaluation (Reviewers dRaU, Bek9, squ7, xjDT)
We have significantly broadened the scope of our evaluation to include modern, state-of-the-art face recognition backbones.
* _New Models_: We added results for ArcFace and MagFace (MobileFaceNet backbones) in Section 6.1 (Table 1 and Figure 2).
* _Defense Generalization_: We included a full evaluation of our defense (StyleAT) on the RepVGG backbone in Section 6.2 (Table 2) and Appendix E, demonstrating that our robustness gains generalize across diverse architectures.
### Methodological Diagram (Reviewers squ7, xjDT)
To address concerns regarding the clarity of the StyleAT training pipeline, we have added a comprehensive architectural diagram in Section 4.2 (Figure 1).
### User Study (Reviewers dRaU, xjDT)
We addressed the need for quantitative perceptual validation by conducting a user study. The results, detailed in Appendix B, confirm that identity is preserved at lower attack budgets but degrades smoothly as the budget increases, validating our choice of parameters in our experiments (Sections 5-6).
### Imperceptible perturbations in DiffPrivate (Reviewers dRaU, xjDT)
We provide further evidence that the baseline (DiffPrivate) relies on imperceptible noise rather than purely semantic edits. Supplementing our previous findings (sensitivity to JPEG compression and pixel-space defenses), we added a spectral analysis in Appendix D. The results show that BoundStyle's energy is concentrated in low frequencies, whereas DiffPrivate relies heavily on high frequencies, further evidencing the presence of non-semantic artifacts.
### Quantitative Semantic Analysis (Reviewer xjDT)
To understand how BoundStyle manipulates interpretable semantic attributes, we performed a Principal Component Analysis (PCA) on the attack vectors in the latent space. The visualization in Appendix F demonstrates that the top principal components of the attack space correspond to coherent semantic factors, such as aging effects, lighting and nose shape modifications, and pose changes.
### Additional Ablation Studies (Reviewers dRaU, squ7, xjDT)
To justify our choice of parameters, we added new results from ablation experiments to Appendix C:
* _Attack Iterations_: We validate our choice of iteration counts, showing that attack success saturates at 30 iterations for BoundStyle and 70 for DiffPrivate, ensuring a fair comparison.
* _Attacks Time and Success Trade-offs_: We include a wall-clock comparison showing that under matched time constraints (6 seconds), BoundStyle effectively degrades robust accuracy (down to 50.9%), whereas DiffPrivate fails to converge (98.6% accuracy), highlighting BoundStyle's superior suitability for adversarial training.

We would like to thank the reviewers again for their valuable feedback. By revising the paper to address the concerns raised during the reviewing process, our paper has become significantly stronger.

---

### Meta-Review · Area_Chair_wE36 · 2026-01-13

**Summary:**

This paper addresses an important problem—semantic adversarial attacks and defenses for face recognition—and proposes an efficient latent-space attack (BOUNDSTYLE) alongside an adversarial training scheme (STYLEAT). While one reviewer finds the attack–defense pairing practical and the efficiency gains compelling, the majority of reviews raise substantial concerns that prevent acceptance.

Across three reviews, the paper is criticized for limited novelty, unclear methodology, and insufficient empirical support for key claims, particularly regarding semantic fidelity and identity preservation. The evaluation is restricted to small, aging datasets and mostly classic backbones, omitting modern face recognition benchmarks and state-of-the-art models. Comparisons to prior attacks are viewed as potentially unfair, and robustness claims rely on incomplete metrics without statistical testing or perceptual validation. Overall, while the direction is promising, the current evidence and rigor are insufficient to support the paper’s broad claims. Given the preponderance of negative reviews and the unresolved methodological and evaluation concerns, I recommend rejection.

**Reviewer Concerns:**

Two reviewers still felt the questions were not well solved.

Two reviewers did not provide a response.

**Reviewer Scores:**

none

---

### Decision · Program_Chairs · 2026-01-26

Reject